# Impacts of climate change, population growth, and power sector decarbonization on urban building energy use

**Chenghao Wang** [1,2,3] ✉, **Jiyun Song** [4,5,6], **Dachuan Shi**[4], **Janet L. Reyna** [7], **Henry Horsey**[7], **Sarah Feron**[8,9], **Yuyu Zhou** [5,10], **Zutao Ouyang** [1], **Ying Li** [11,12] & **Robert B. Jackson** [1,13,14]

Climate, technologies, and socio-economic changes will influence future building energy use in cities. However, current low-resolution regional and state-level analyses are insufficient to reliably assist city-level decision-making. Here we estimate mid-century hourly building energy consumption in 277 U.S. urban areas using a bottom-up approach. The projected future climate change results in heterogeneous changes in energy use intensity (EUI) among urban areas, particularly under higher warming scenarios, with on average 10.1–37.7% increases in the frequency of peak building electricity EUI but over 110% increases in some cities. For each 1 °C of warming, the mean city-scale space-conditioning EUI experiences an average increase/decrease of ~14%/~10% for space cooling/heating. Heterogeneous city-scale building source energy use changes are primarily driven by population and power sector changes, on average ranging from −9% to 40% with consistent south–north gradients under different scenarios. Across the scenarios considered here, the changes in city-scale building source energy use, when averaged over all urban areas, are as follows: −2.5% to −2.0% due to climate change, 7.3% to 52.2% due to population growth, and −17.1% to −8.9% due to power sector decarbonization. Our findings underscore the necessity of considering intercity heterogeneity when developing sustainable and resilient urban energy systems.

Residential and commercial buildings together are responsible for 39% of the U.S. energy consumption and 28% of the U.S. greenhouse gas emissions[1]. In densely populated urban areas, the share of energy use and emissions attributable to buildings can be even higher[2]. With

ongoing climate change, technological innovations, and socio-economic developments, future urban building energy use in the U.S. is expected to change as well[3,4]. However, the current methods and data used for predicting future building energy consumption in the

[1]Department of Earth System Science, Stanford University, Stanford, CA, USA. [2]School of Meteorology, University of Oklahoma, Norman, OK, USA. [3]Department of Geography and Environmental Sustainability, University of Oklahoma, Norman, OK, USA. [4]Department of Mechanical Engineering, The University of Hong Kong, Pokfulam Road, Hong Kong SAR, China. [5]Department of Geography, The University of Hong Kong, Pokfulam Road, Hong Kong SAR, China. [6]State Key Laboratory of Water Resources Engineering and Management, Wuhan University, Wuhan, China. [7]National Renewable Energy Laboratory, Golden, CO, USA. [8]Universidad de Santiago de Chile, Santiago, Chile. [9]University of Groningen, Groningen, The Netherlands. [10]Institute for Climate and Carbon Neutrality, The University of Hong Kong, Pokfulam Road, Hong Kong SAR, China. [11]Engineering Research Center of Eco-environment in Three Gorges Reservoir Region, Yichang, China. [12]College of Hydraulic and Environmental Engineering, China Three Gorges University, Yichang, China. [13]Woods Institute for the Environment, Stanford University, Stanford, CA, USA. [14]Precourt Institute for Energy, Stanford University, Stanford, CA, USA. ✉e-mail: chenghao.wang@ou.edu

U.S. suffer from limited resolutions[5–7], leading to substantial uncertainties in existing projections. Consequently, these projections often lack the necessary detail to effectively inform decision-making in individual cities.

Current understanding of future building energy use mainly draws upon top-down and bottom-up studies[8]. Compared with top-down approaches, bottom-up methods can have finer spatial resolutions that are more suitable for subnational analysis[7,9]. Statistical bottom-up models project future building energy demand based on mathematical relationships between historical energy use and weather data (usually temperature or temperature-related variables, such as cooling and heating degree days)[5,6]. Nevertheless, considering the lack of high-resolution energy consumption data and/or meteorological records, statistics-based city-scale studies have been rare[10,11]. In comparison, engineering-based bottom-up approaches can achieve a higher spatial resolution and more explicitly account for the impacts of future changes on specific end-use types[8,12]. Leveraging hourly or even sub-hourly physics-based building energy models driven by meteorological data, engineering-based approaches also enable detailed peak demand analysis, providing a more comprehensive understanding of energy demand patterns.

Meteorological data are one of the essential factors that determine building energy consumption and its evolution over time[13]. However, bottom-up methods, especially engineering-based ones, often oversimplify the impacts of future climate changes on meteorological conditions. Due to the high computational cost and lack of reliable, long-term meteorological data, building energy models have relied primarily on typical meteorological year (TMY) data that piece together "typical" historical months of weather observations from different years. Future weather patterns are then approximated with a "morphing method", which imposes monthly or daily changes in meteorological variables from climate projections onto historical TMY data[12,14]. These TMY-based methods cannot capture the interannual variability of meteorological conditions, particularly during extreme events, nor can they adequately reflect the interdependence among changes in meteorological variables under future climate conditions[15].

In addition to meteorological conditions, urban building energy use is influenced by a range of factors, including socio-economic developments, building technology advancements, possible changes in individual behavior, and evolutions in the electric power sector. As a key indicator of socio-economic developments[16], urban population growth directly impacts total floor area and consequently drives changes in city-scale building energy use[4]. Household income and electricity price are additional socio-economic factors that can influence urban building energy use, especially cooling demand in low- and middle-income countries with low air conditioning (AC) adoption[17,18]. Building energy use can be further affected by shifts in building technologies, including electrification and energy-efficiency measures, although their impacts are contingent on occupant behavior such as thermostat setpoints[19,20]. Moreover, the primary energy consumption of buildings (source energy), including electricity and fossil fuels consumed in buildings (final or site energy) and energy losses during generation, transmission, and distribution, is influenced by electric power sector evolutions. In particular, the transition from fossil fuels to renewables in the power sector can substantially reduce energy losses during electricity generation[21]. Despite their importance, these various possible futures and their spatial variability have rarely been considered in existing bottom-up studies for cities, and their impacts on urban building energy use have not been well characterized.

In this study, we quantify the city-scale building energy use for 277 urban areas across the contiguous U.S. (CONUS) in the mid-21st century, with a focus on assessing the impacts of climate change, population growth, and power sector decarbonization. We develop a hybrid bottom-up modeling approach that integrates extensive hourly physics-based building energy simulations with statistical models

constructed using a high-quality and high-resolution end-use load database[22]. Decadal-scale hourly meteorological data derived from multiple sources for historical and future periods (the 2010s and 2050s) enable a multiscale, detailed assessment of climate change impacts on urban building energy use. Specifically, we downscale and bias-correct climate projections from a set of Coupled Model Intercomparison Project Phase 6 (CMIP6) climate models under four warming scenarios. We maintain building technologies, occupant behavior, and other socio-economic factors at their current levels, except for the AC penetration rate, and establish baseline projections for different future warming scenarios.

For site energy use of buildings (hereafter simply "energy use", unless otherwise specified), we focus on energy use intensity (EUI), an indicator calculated as the final energy consumed in buildings divided by the total floor area of all buildings in an urban area. The use of building EUI can effectively isolate the impact of climate change from that of population change. Based on the types of final energy consumed in buildings, building EUI can be classified as electricity EUI (E-EUI) and non-electricity EUI (NE-EUI, including natural gas, fuel oil, and propane EUIs). For source energy use of buildings in each urban area, we incorporate high-resolution population projections consistent with different warming scenarios, and further consider the potential impact of two possible electric power sector trajectories with different levels of decarbonization.

## Results
### Impacts of climate change on annual electricity energy use intensity
Four representative warming scenarios (SSPX-Y; see Methods) are considered in this study: SSP1-2.6, SSP2-4.5, SSP3-7.0, and SSP5-8.5. These scenarios encompass a diverse set of socio-economic storylines and global radiative forcing levels, representing a wide range of possible future climates. Under different warming scenarios, the mid-century annual building E-EUI for cooling in U.S. urban areas, on average, is projected to increase by 11.9–25.6% when compared with the conditions during the period 2010–2019 (hereafter reference decade) (Supplementary Fig. 1). In comparison, the mean mid-century annual building E-EUI for space heating is projected to decrease by 13.5–21.2% when compared with the reference-decade level (Supplementary Fig. 2). However, the change in annual total E-EUI, which includes space conditioning, water heating, and all other miscellaneous electric end-use types, varies substantially among urban areas across different warming scenarios (Fig. 1). The largest increases in annual E-EUI mainly occur in the South, Southwest, West, and Southeast (see division in Supplementary Fig. 3), with the maximum relative increase ranging from 3.2% under SSP1-2.6 to 7.2% under SSP5-8.5. Most urban areas with annual E-EUI declines are in the northern part of the country, especially the Northwest and Northern Rockies and Plains, with a reduction of up to 4.0% under the high warming scenario SSP5-8.5.

The net change in building E-EUI for each urban area hinges on whether the projected increase in cooling demand can be partially or entirely offset by the decline in space heating need in a warmer future. Therefore, the spatial heterogeneity in the projected changes is driven not only by regional disparities in future warming, but also by variations in the share of electricity consumption for cooling and space heating (Supplementary Figs. 4, 5). The latter factor leads to a diverse range of directions for annual E-EUI changes among urban areas within the same state and even within the same county (e.g., urban areas in California and Ohio; Fig. 1), which has been largely overlooked by previous regional and country-level studies[3,5,7,23,24].

### Impacts of climate change on seasonal and hourly electricity energy use intensity
The high-granularity modeling approach developed in this study (Methods) allows for the examination of mid-century city-scale

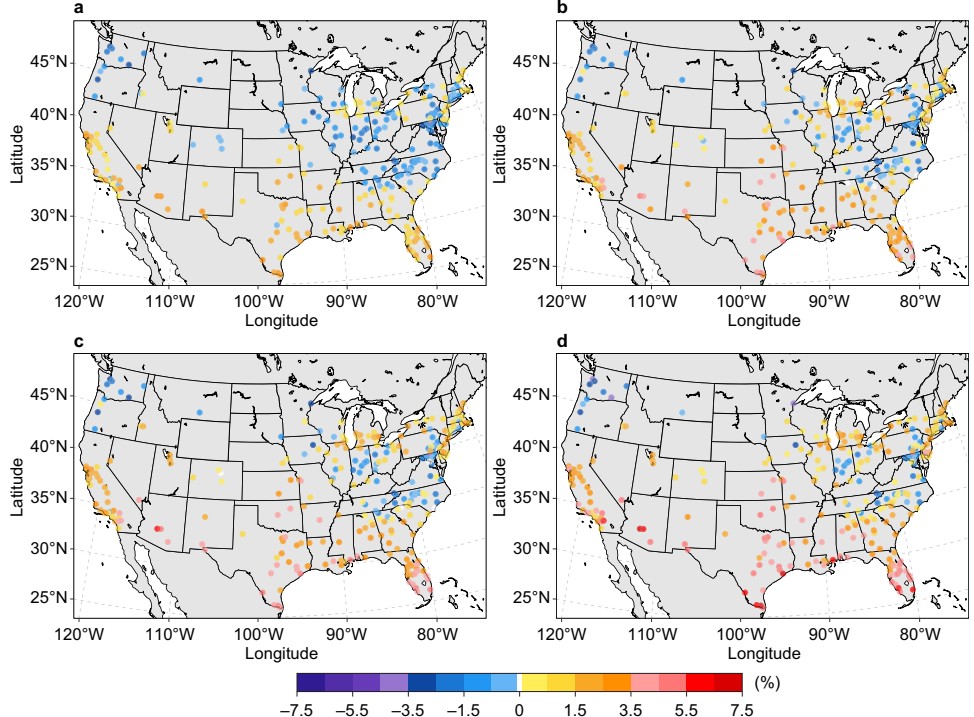

**Fig. 1 | Change in annual electricity energy use intensity in the 2050s relative to the reference decade under four illustrative emissions and concentration scenarios with different global warming levels. a** SSP1-2.6 scenario. **b** SSP2-4.5 scenario. **c** SSP3-7.0 scenario. **d** SSP5-8.5 scenario. Each point represents the relative change (%) based on the ensemble mean of the simulations driven by 10 CMIP6 models under each SSPX-Y scenario. Sources of base map: U.S. Census Bureau and Natural Earth. Source data are provided as a Source Data file.

building EUI and its frequency across different temporal scales. Notably, large seasonal variations are observed in the projected building E-EUI changes. Compared with the historical reference period, the decreases and increases in future E-EUI primarily occur in November–March and May–September, respectively (Supplementary Fig. 6). These variations combined with the historical load profiles ultimately determine the net change on the annual scale (Fig. 1). Substantial increases in E-EUI during the warm season (May–September) and the top 5% hottest days are projected for all climate regions, and in both cases, the changes are much larger than the annual average change (Fig. 2). This is attributable to the rise in AC adoption[17] (Methods) as well as the growing demand for cooling in response to future warming. For example, the projected increase in the future AC saturation rate for urban areas in the Northwest nearly doubles the increase in building E-EUI for space cooling under all four warming scenarios (Supplementary Fig. 3). As a result, urban areas in the Northwest see an average increase of 13.6% in E-EUI on the hottest days under SSP5-8.5, which is more than twice the warm season counterpart (6.1%) (Fig. 2).

Over the diurnal cycle, the most substantial decrease in building E-EUI occurs in the early morning (~5:00–7:00 local time; dependent on region and scenario), mainly driven by reduced space heating demand (Supplementary Figs. 7–9). In comparison, the maximum increase in building E-EUI occurs in the early afternoon (~13:00–15:00 local time), as determined by the elevated space cooling demand. To examine the change in peak demand frequency, we define peak hours as hours with E-EUI higher than the current (the reference decade) 95th percentile. On average, the spatial pattern of the relative change in peak demand frequency (number of peak hours) under different warming scenarios resembles that of the relative annual mean changes (Supplementary Fig. 10; cf. Fig. 1), but with considerably larger magnitudes. For instance, the maximum city-scale peak demand frequency increase ranges from 94% to 170% across different warming scenarios

(i.e., ~2–3 times the number of historical peak hours). In particular, several coastal urban areas in California are projected to experience the largest increase in peak demand frequency (e.g., >110% under SSP5-8.5). Given that the capacity requirement of the electric grid is largely determined by the peak demand, these changes suggest a considerable demand pressure on the electric grid and an increasing need for grid resilience, especially during heat waves.

## Impacts of climate change on non-electricity energy use intensity

Fossil fuels, including natural gas, fuel oil, and propane, account for ~40% of the building energy use in CONUS urban areas. These fuels are utilized for various purposes, including space heating, water heating, and other miscellaneous end-use categories. Similar to E-EUI for space heating, NE-EUI for space heating generally diminishes with climate change, resulting in reduced total NE-EUI under all four warming scenarios (Fig. 3 and Supplementary Figs. 11–14). Mid-century changes in city-scale natural gas EUI (Supplementary Fig. 11) closely align with those for total NE-EUI, whereas changes in fuel oil and propane EUIs are geographically more heterogeneous (Supplementary Figs. 12, 13). This difference is caused by the more diverse space-heating share in fuel oil and propane consumption among urban areas. When averaged for all urban areas, the changes in mid-century total NE-EUI relative to the reference-decade level are −7.7% and −11.6% under the low (SSP1-2.6) and high (SSP5-8.5) warming scenarios, respectively. Note that we do not explicitly model building technology change other than increased AC adoption, so this is a product of warming-induced heating demand change, rather than efficiency improvement or fuel switching.

Space heating dominates building fossil fuel use for urban areas in the Upper Midwest and Ohio Valley. For example, space heating on average accounts for 73.3–74.5% of mid-century total natural gas consumed in buildings across four SSPX-Y scenarios in these two climate regions (Supplementary Fig. 15; cf. 77.4% during the reference

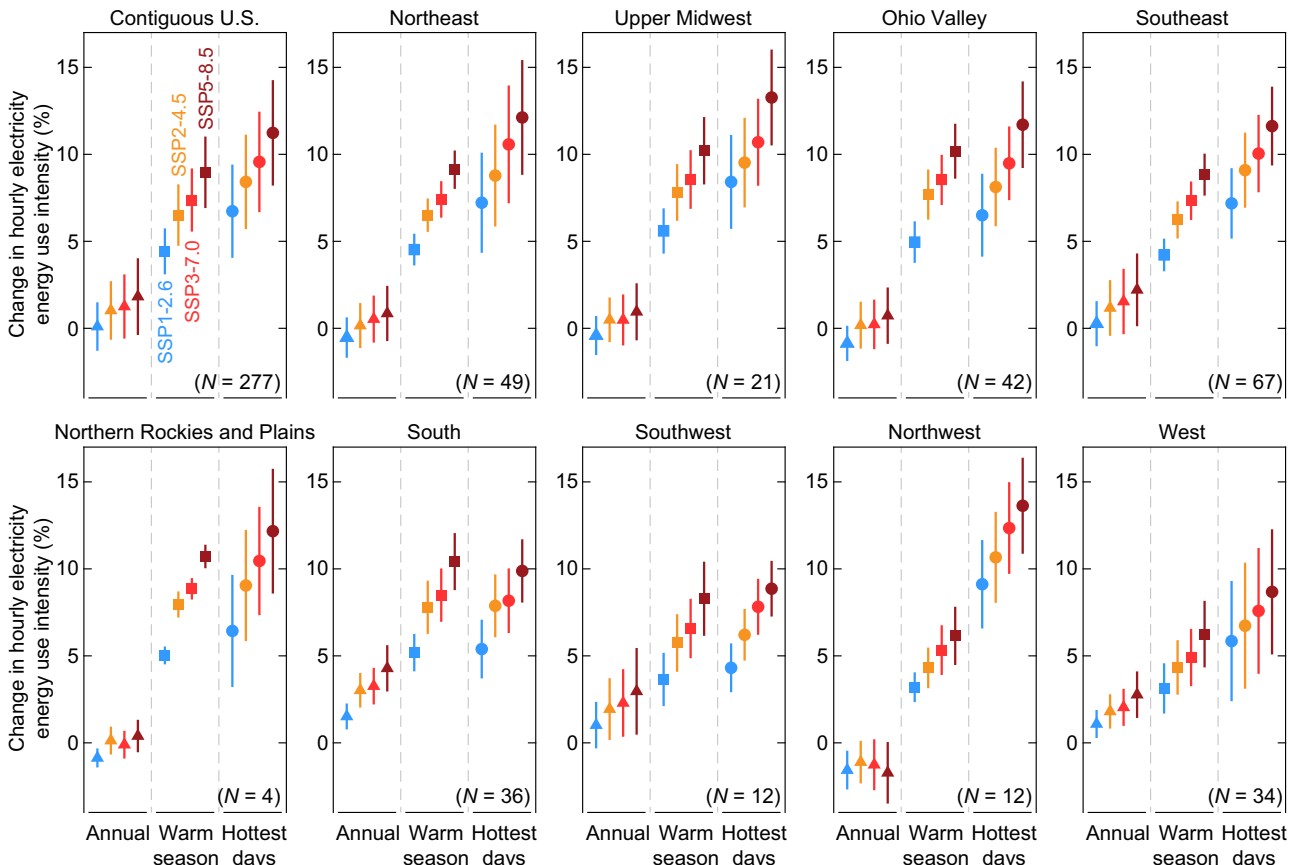

**Fig. 2 | National and regional changes in hourly electricity energy use intensity in the 2050s relative to the reference decade when averaged over the annual scale, the warm season, and the top 5% hottest days.** In each subplot, the left, middle, and right panels show changes in the annual average, the average of the warm season (May–September), and the average of the top 5% hottest days over the simulated decade, respectively, based on the ensemble mean of the simulations driven by 10 CMIP6 models under each SSPX-Y scenario. Error bars represent the variability (±1 standard deviation) among urban areas within each region. *N* in each subplot denotes the number of urban areas. The division of nine climate regions is shown in Supplementary Fig. 3. Source data are provided as a Source Data file.

decade). Therefore, urban areas in these regions are projected to have the most substantial reduction in annual total NE-EUI in a warmer future, with the maximum reduction ranging from 15.1% to 19.2% (Fig. 3). In comparison, urban areas in the southern part of the country exhibit much smaller changes in total NE-EUI (Fig. 3). For instance, despite the considerable changes in NE-EUI for space heating (Supplementary Fig. 14), the change in total NE-EUI is expected to be less than 1.5% for most urban areas in Florida. Overall, non-space heating fossil fuels are projected to account for an increasing share of city-scale building fossil fuel consumption in the mid-century, barring major changes in the fuel demand of non-space heating appliances, such as through end-use electrification policies.

### Impacts of climate change on total energy use intensity
Under all four SSPX-Y scenarios, the total EUI of buildings (the sum of E-EUI and NE-EUI) is projected to drop in most U.S. urban areas in the 2050s, driven by the reduced space heating demand (Supplementary Fig. 16). The reduction in building natural gas consumption on average accounts for ~88% of the reduced NE-EUI. Urban areas with a projected increase in total EUI are predominantly located in the southern part of the country, including the Southeast, South, West, and Arizona. The changes in cooling and space heating demands intensify with future warming, resulting in greater spatial heterogeneity under warmer scenarios. For example, city-scale changes in total EUI range from −10.4% to 2.5% under the low warming scenario SSP1-2.6, but from −13.0% to 6.1% under the high warming scenario SSP5-8.5.

### City-scale response of energy use intensity for space-conditioning to warming
Unlike the observed monotonic increase in electricity use and EUI with global warming levels in end-of-century studies[3,25], the conditions during the mid-century show a high degree of heterogeneity. Specifically, over 18% of the U.S. urban areas are projected to have non-monotonic changes in E-EUI with mid-century warming levels. For example, the smallest changes in annual E-EUI are projected under SSP2-4.5 for most urban areas in the Northwest (Fig. 2; e.g., Seattle, Washington). In contrast to these nonmonotonic changes, building EUIs for cooling and space heating both exhibit statistically significant linear dependence on mid-century warming levels, as suggested by linear regression models constructed using a set of 40 projections for each urban area (Methods).

On average, city-scale building E-EUI for cooling rises by 13.8% for each 1 °C of warming in the urban area (Fig. 4a). The greatest sensitivity to temperature change (up to 34.9%/°C) is projected for urban areas in the Northwest, coastal California, Upper Midwest, and Northeast, where the share of electricity for cooling is low. City-scale building E-EUI for space heating on average drops by 11.0% for each 1 °C of urban warming (Fig. 4b) and is less influenced by the share of electricity for heating. The most sensitive urban areas include those in the Southeast, South, and coastal California. In addition, all three types of NE-EUIs for space heating exhibit patterns of warming responses that are fairly similar to their electricity counterpart. The average reductions per 1 °C of warming in the urban area are 10.5%, 8.7%, and 10.2%

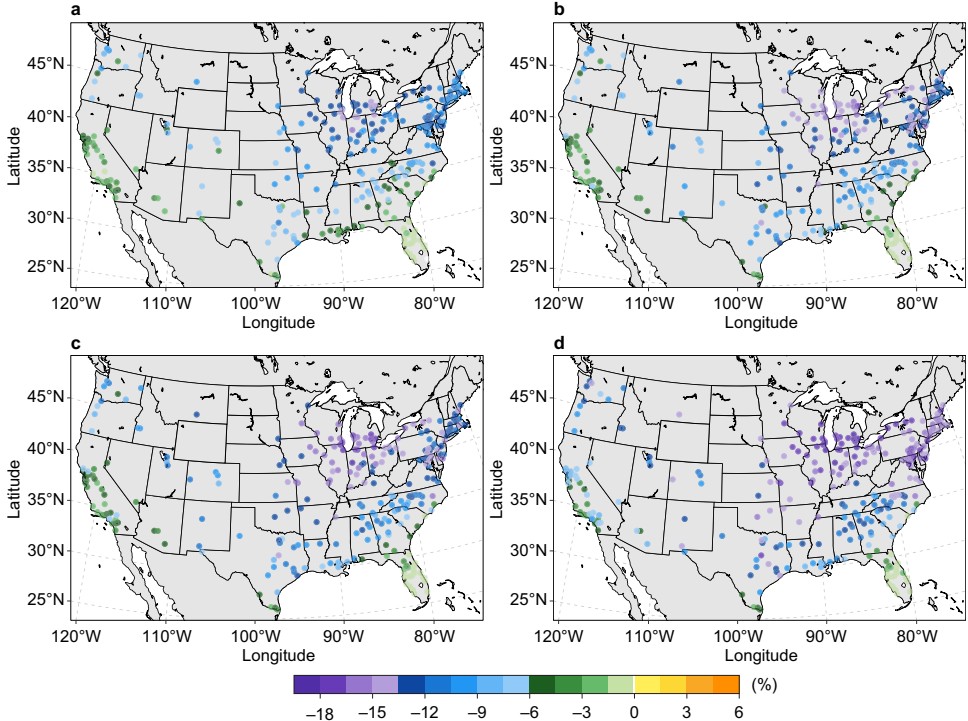

**Fig. 3 | Change in annual non-electricity (fossil fuel) energy use intensity in the 2050s relative to the reference decade under four illustrative emissions and concentration scenarios with different global warming levels. a** SSP1-2.6 scenario. **b** SSP2-4.5 scenario. **c** SSP3-7.0 scenario. **d** SSP5-8.5 scenario. Types of fossil fuels consumed by buildings include natural gas, fuel oil, and propane. Each point represents the relative change (%) based on the ensemble mean of the simulations driven by 10 CMIP6 models under each SSPX-Y scenario. Results for changes in annual natural gas energy use intensity, annual fuel oil energy use intensity, and annual propane energy use intensity are shown in Supplementary Figs. 11–13. Sources of base map: U.S. Census Bureau and Natural Earth. Source data are provided as a Source Data file.

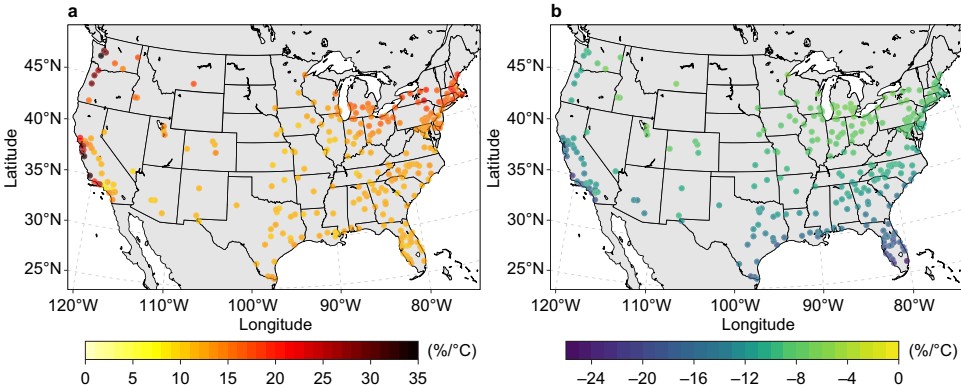

**Fig. 4 | Response of annual electricity energy use intensity for space conditioning to a 1 °C temperature change in the 2050s relative to the reference decade. a** Response of annual electricity energy use intensity for cooling. **b** Response of annual electricity energy use intensity for space heating. Each point represents the response of energy use intensity to warming evaluated as the slope (%/°C) of the linear regression fit to all simulations under all SSPX-Y scenarios (see Methods). Results for the response of annual natural gas energy use intensity for space heating are shown in Supplementary Fig. 17b. Sources of base map: U.S. Census Bureau and Natural Earth. Source data are provided as a Source Data file.

for natural gas (Supplementary Fig. 17b), fuel oil, and propane, respectively.

## Impacts of future changes on total source energy use

In addition to climate change, mid-century urban building source energy consumption in the U.S. is also influenced by population dynamics and electric power sector decarbonization. Urban population growth directly affects the total source energy consumption of buildings through changes in total floor area. The decarbonization process of the electric power sector mainly impacts primary energy consumption during electricity generation before delivering to buildings. On the one hand, switching from coal-based power plants to renewable energy technologies reduces both greenhouse gas emissions and primary energy losses (Supplementary Table 9). On the other hand, the implementation of carbon capture and storage technologies can result in efficiency reductions and energy penalties[26,27]. Here we consider two possible futures of the U.S. electric power sector, and assess building source energy consumption through a set of site-to-source conversion factors (Methods; Fig. 5).

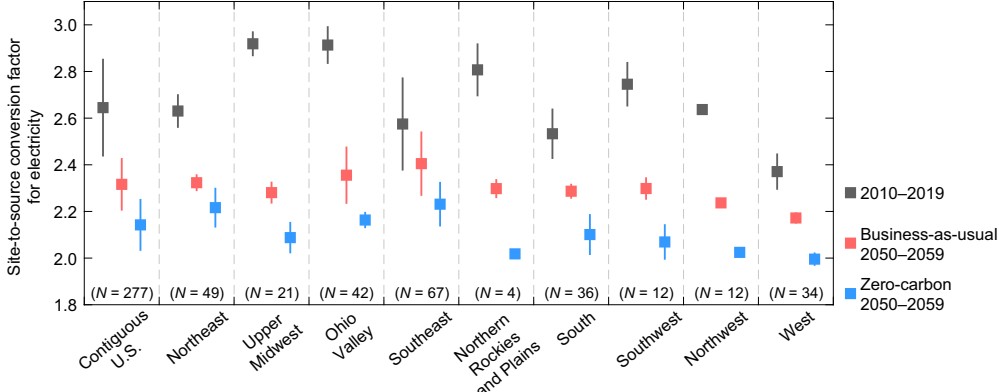

**Fig. 5 | National and regional site-to-source conversion factors for electricity in the 2010s and 2050s with electric power sector decarbonization under the business-as-usual and zero-carbon scenarios.** Data are presented as mean value, with error bars representing the variability (±1 standard deviation) among urban areas within each region. Conversion factors in the 2050s for each city are averaged across four SSPX-Y scenarios. N denotes the number of urban areas in each region. Source data are provided as a Source Data file.

The business-as-usual power sector scenario represents a continued evolution of current energy technologies and policies but assumes no new carbon policies in the future. With the current state and federal $CO_2$ emission policies and nominal technology cost projections, the national share of coal-fired generation is expected to decline from ~26% to ~6% in the 2050s (Supplementary Fig. 18). As a result, the mean city-scale site-to-source conversion factor for electricity decreases by 12.4% compared with the reference decade (Fig. 5), suggesting an overall nationwide improvement in electricity generation efficiency. Different from the business-as-usual scenario, the zero-carbon scenario assumes rapid decarbonization and net-zero carbon emissions by the mid-century. This scenario includes more stringent emission requirements, resulting in a complete phase-out of coal-fired power plants projected to occur around 2035 (Supplementary Fig. 18). Meanwhile, the national share of renewables in mid-century electricity generation is expected to surpass 70% (cf. ~50% under the business-as-usual scenario). On average, the city-scale site-to-source conversion factor for electricity in the 2050s under the zero-carbon scenario is 19.0% lower than in the reference decade, with the most substantial reductions in urban areas located in the Upper Midwest, Northern Rockies and Plains, Ohio Valley, and Southwest (Fig. 5).

For both power sector decarbonization trajectories, the increase in source energy consumption for U.S. urban areas is projected to peak under SSP5-8.5, higher than those under SSP1-2.6 and SSP2-4.5, as climate change and population growth impacts offset source energy efficiency gains. The average city-scale source energy consumption changes under SSP5-8.5 are 39.8% and 31.8% for business-as-usual scenario and zero-carbon scenario, respectively. In contrast, decreasing source energy use is projected for most urban areas under SSP3-7.0, with an average change of −3.5% for a business-as-usual power sector and −9.0% for a zero-carbon power sector (Figs. 6, 7). Compared with the business-as-usual scenario, a zero-carbon power sector results in relatively lower building source energy use across all warming scenarios. For example, with mid-century climate change under SSP3-7.0, more urban areas are expected to experience increased building source energy use in the business-as-usual scenario (33.2%) compared with the zero-carbon scenario (7.6%). In addition, consistent south–north gradients of source energy consumption changes are projected in U.S. urban areas for both power sector futures (Figs. 6, 7). Furthermore, substantial within-region variations of city-scale source energy use changes are projected in several regions, especially for urban areas in Michigan, California, and Florida under SSP3-7.0. These variations are largely determined by heterogeneous changes in future population and warming-induced space-conditioning demand.

## Attributions of source energy use change

We employ the logarithmic mean Divisia index method[28] to quantify the relative contributions of climate change, population growth, and power sector decarbonization to the projected mid-century changes in building source energy consumption (Methods). As revealed by the decomposition, under all SSPX-Y scenarios except SSP3-7.0, population growth serves as the primary driver of future source energy use changes for most urban areas, followed by power sector change and climate change (Fig. 8 and Supplementary Figs. 19–22). Urban population growth, owing to socio-economic developments, contributes to an average increase of 31.5%, 29.0%, 7.3%, and 52.2% in city-scale source energy use under SSP1-2.6, SSP2-4.5, SSP3-7.0, and SSP5-8.5, respectively. These relative changes are in general consistent with the projected city-scale population changes (Supplementary Fig. 23).

With the gradually decreasing share of fossil fuels and rising share of renewables under the business-as-usual scenario, power sector decarbonization reduces future source energy use in all urban areas (Fig. 8a–d and Supplementary Fig. 21). The average reduction in the 2050s attributable to the business-as-usual power sector ranges from 8.9% under SSP3-7.0 to 10.8% under SSP5-8.5. Urban areas in the Northeast, Southeast, and West are projected to have the smallest contributions from power sector changes, primarily due to the already low share of coal in current electricity generation (Supplementary Table 1). For example, under the business-as-usual scenario, the mean reduction in city-scale source energy consumption due to power sector evolution ranges from 6.7% to 8.1% in the West (cf. 13.1–15.9% in the Ohio Valley). Note that the relative change in building source energy use induced by power sector decarbonization is smaller than the change in conversion factors for electricity, as source energy also includes fossil fuels delivered to buildings.

More aggressive power sector decarbonization, with the complete phase-out of coal use, could further compensate for the population-induced increase in source energy consumption (Fig. 8e–h and Supplementary Fig. 22). The average reduction of mid-century source energy use in U.S. urban areas attributable to net-zero power sector policies varies from 14.2% under SSP3-7.0 to 17.1% under SSP5-8.5. The contribution of power sector decarbonization varies across different climate regions, partly resulting from historical dependence on fossil fuels in electricity generation. For example, many urban areas in Texas and Florida that heavily rely on fossil fuels to generate electricity are projected to transition toward renewables-dominant generation. Meanwhile, the remaining fossil fuel-fired generation under the business-as-usual scenario is also expected to be largely replaced by renewables under the zero-carbon scenario (Supplementary Table 1). Therefore, the contribution of power sector decarbonization

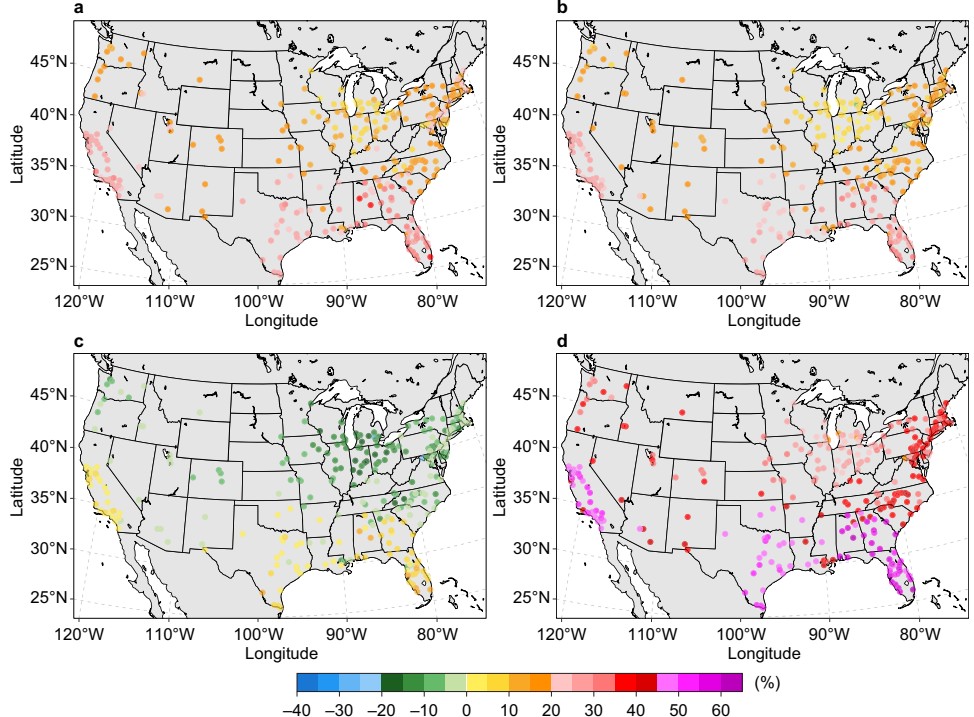

**Fig. 6 | Change in annual source energy consumption in the 2050s relative to the reference decade under four illustrative emissions and concentration scenarios influenced by electric power sector decarbonization under the business-as-usual scenario. a** SSP1-2.6 scenario. **b** SSP2-4.5 scenario. **c** SSP3-7.0 scenario. **d** SSP5-8.5 scenario. Each point represents the relative change (%) based on the ensemble mean of the simulations driven by 10 CMIP6 models under each SSPX-Y scenario. Sources of base map: U.S. Census Bureau and Natural Earth. Source data are provided as a Source Data file.

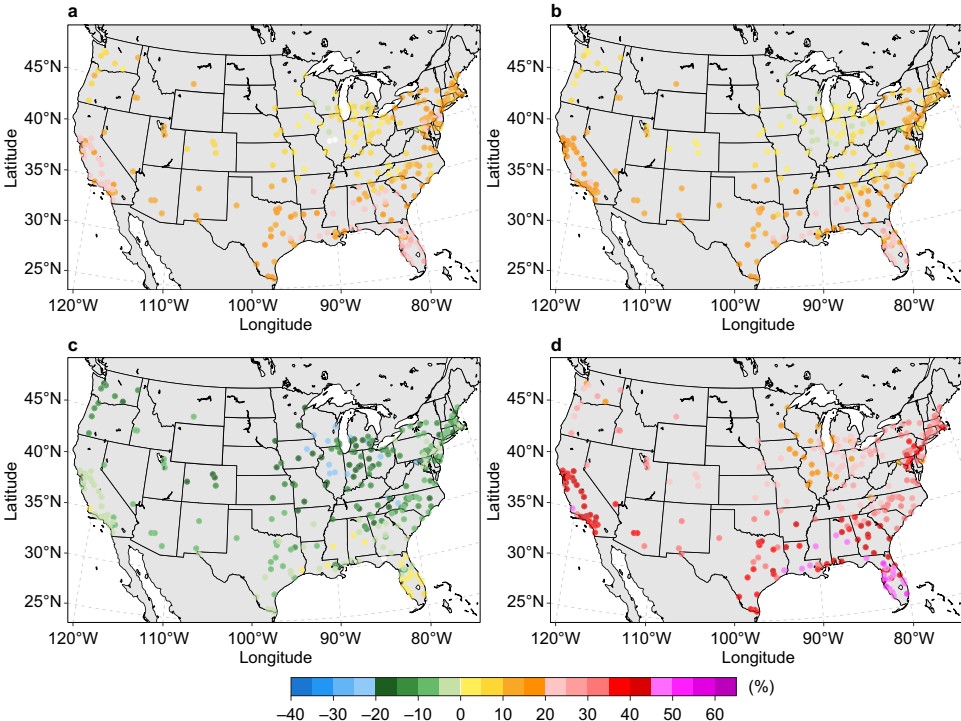

**Fig. 7 | Change in annual source energy consumption in the 2050s relative to the reference decade under four illustrative emissions and concentration scenarios influenced by electric power sector decarbonization under the zero-carbon scenario. a** SSP1-2.6 scenario. **b** SSP2-4.5 scenario. **c** SSP3-7.0 scenario. **d** SSP5-8.5 scenario. Each point represents the relative change (%) based on the ensemble mean of the simulations driven by 10 CMIP6 models under each SSPX-Y scenario. Sources of base map: U.S. Census Bureau and Natural Earth. Source data are provided as a Source Data file.

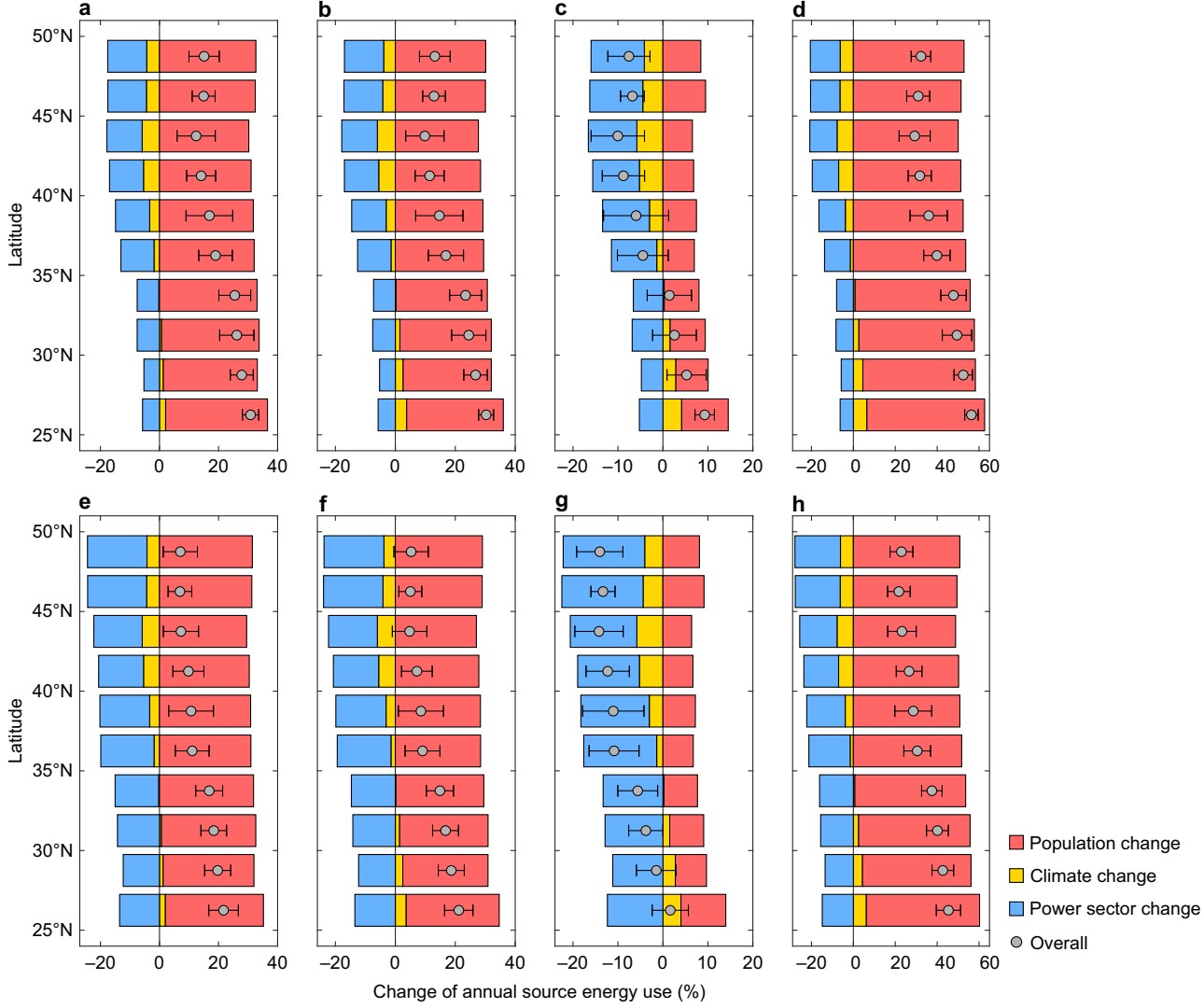

**Fig. 8 | Latitudinal profiles of changes in annual source energy consumption in the 2050s relative to the reference decade attributed to climate change, population change, and electric power sector decarbonization. a** SSP1-2.6 with business-as-usual scenario. **b** SSP2-4.5 with business-as-usual scenario. **c** SSP3-7.0 with business-as-usual scenario. **d** SSP5-8.5 with business-as-usual scenario. **e** SSP1-2.6 with the zero-carbon scenario. **f** SSP2-4.5 with the zero-carbon scenario. **g** SSP3-7.0 with the zero-carbon scenario. **h** SSP5-8.5 with the zero-carbon scenario. The overall effect is the change in annual source energy consumption, which equals the sum of contributions from three drivers. Here the overall effect averaged across all urban areas within latitude band is presented, with error bars representing the variability (±1 standard deviation) among urban areas. Results for the entire U.S. and nine climate regions are shown in Supplementary Figs. 21, 22. Source data are provided as a Source Data file.

in a zero-carbon future is approximately twice that in a business-as-usual future in these regions (Supplementary Figs. 21, 22).

When averaged across all urban areas, climate change is projected to reduce city-scale building source energy use by 2.5%, 2.1%, 2.0%, and 2.4% under SSP1-2.6, SSP2-4.5, SSP3-7.0, and SSP5-8.5, respectively (Supplementary Figs. 19, 20). The relatively small mean contribution of climate change, compared with that of population growth or power sector decarbonization, is partly because global warming causes total building energy use to increase in some urban areas while decreasing in others (Fig. 8). In fact, under different warming scenarios, the maximum climate change-induced increase and decrease in city-scale source energy consumption are projected to be around 3–8% and 8–11%, respectively. Notably, under SSP3-7.0, the contribution of climate change is of the same order of magnitude as that of population change for some urban areas in the Northeast, Upper Midwest, and Ohio Valley, where the increase in source energy use caused by population growth can be largely or even completely offset by the warming-induced reduction.

The south–north contrast of the changes in city-scale building source energy consumption is jointly determined by the impacts of climate change and power sector decarbonization (Fig. 8). With the projected warming, the increase in cooling demand is expected to control the shift in building source energy use for urban areas in the southern regions, whereas the declining space heating demand becomes the dominant factor for those in the northern regions. In addition, the transition toward renewables proves to be more effective in reducing primary energy losses in several northern regions with a higher share of fossil fuel-fired electricity generation, such as the Upper Midwest and Ohio Valley. These power sector decarbonization-induced spatial variations in energy losses further reinforce the south–north gradient of mid-century changes in city-scale building source energy consumption.

## Discussion
The high spatial resolution of our approach reveals heterogeneous city-scale building EUI changes in response to future warming

(Supplementary Fig. 16). Depending on building stock characteristics, urban areas in the same state or even the same county may experience varying future changes in building EUI under different warming scenarios (Figs. 1 and 3). Consequently, the need for future urban energy infrastructure deployment will differ among urban areas. For instance, considering population changes alone when making decisions on electric grid expansion in urban areas with similar population growth rates but distinct EUI changes (e.g., California) may lead to underestimating or overestimating actual electricity demand. Therefore, the planning of electricity transmission, distribution, and storage infrastructure for each urban area must factor in these warming-induced variations. Likewise, the spatial variation in NE-EUI changes implies potentially varying requirements for pipelines and compressor stations to supply fossil fuels to buildings. This is especially the case for SSP3-7.0, under which the magnitude of population-induced change is similar to warming impacts on building fossil fuel use in many urban areas (Fig. 3 and Supplementary Fig. 23). These intercity variations emphasize the location-dependent challenges and opportunities each urban area will encounter in terms of energy saving and energy efficiency improvement.

The high temporal resolution of our approach enables a detailed assessment across multiple temporal scales. For instance, urban areas in the U.S. are projected to experience diurnal and seasonal changes in total EUI that are much larger than annual mean changes (Fig. 2 and Supplementary Figs. 6–9). These temporally heterogeneous changes should be taken into account when planning for regional generation capacity expansion and energy storage, especially for a future power grid that relies on variable renewables[29]. In addition, the substantial increases in the intensity and frequency of summer peak electricity demand in U.S. urban areas (e.g., Supplementary Fig. 10) could necessitate not only a higher grid capacity but also greater resilience against blackouts during extreme heat waves. Recent studies have highlighted the rapidly elevated risks of heat-related mortality and morbidity during compound heat wave and grid failure events in several U.S. urban areas[30,31]. Given the anticipated rise in urban cooling demand across all warming scenarios (Supplementary Fig. 1), enhancing power grid resilience will become increasingly crucial to mitigate potential health risks and cope with heat extremes.

Previous studies have traditionally used global mean temperature changes to assess the linear or near-linear responses of energy use to future warming[3,25]. In comparison, here we derive city-scale responses of space-conditioning EUI to local air temperature changes. These linear responses are informative for urban planning and policy-making processes, particularly in the development of strategies for urban heat mitigation and adaptation. To improve urban sustainability, cities worldwide are actively developing and implementing measures such as green infrastructure and cool materials to reduce heat stress and building energy use during warm seasons[32,33]. While mesoscale meteorological models can reasonably quantify the city-scale cooling effect of heat mitigation strategies, the evaluation of their effectiveness in reducing building energy use is typically conducted on a case-by-case basis at the building- or community-scale[34–36]. Considering the potential tradeoff between cooling energy savings and space heating penalties as well as distinct building stock characteristics across urban areas, the resulting city-scale energy use change is often uncertain[37–39]. This is partly reflected in some urban areas along coastal California, where cooling and space heating are both highly sensitive to temperature change (Fig. 4). The city-scale responses quantified in this study (Fig. 4 and Supplementary Fig. 17) provide a valuable solution to this issue for urban planners and decision-makers. For instance, when assessing the impact of specific heat mitigation strategies in an urban area, the quantified linear response enables convenient estimation of city-scale changes in space-conditioning building energy use based on predicted temperature changes (e.g., from meteorological models).

Under all warming scenarios except SSP3-7.0, the warming-induced change in source energy consumption for urban buildings is about one order of magnitude smaller than the increase governed by population growth. The projected rise in source energy use under most warming scenarios (Figs. 6–8) indicates potential changes in associated $CO_2$ emissions, signaling the great challenges ahead in achieving decarbonization. Under the business-as-usual scenario, fossil fuel-based power plants continue to emit $CO_2$ when supplying electricity to buildings. However, these emissions can be offset through the implementation of decommissioning and negative emission technologies, such as carbon capture and storage and direct air capture, as seen under the zero-carbon scenario. Yet, with the growing population, direct fossil fuel combustion in buildings is projected to increase in nearly all urban areas under SSP1-2.6, SSP2-4.5, and SSP5-8.5, and in one-third of urban areas under SSP3-7.0 (Supplementary Fig. 24). Correspondingly, their associated $CO_2$ emissions remain a key barrier to achieving decarbonization. Therefore, rapid electrification of future urban buildings is of equal importance to power sector decarbonization[40], particularly under higher warming/emission scenarios. We note this as an important area for future study.

Changes in building construction and operation, building codes, appliance efficiency, and other economic factors such as income and price can also influence building EUI[7,9,24]. For example, a recent study[41] demonstrated that aggressive improvements in building efficiency, such as deploying higher efficiency appliances and space-conditioning equipment, higher performance building envelope, and smart thermostats, can potentially reduce electricity consumption in U.S. buildings by up to one quarter in 2050, assuming no changes in climate. The same study also estimated that full electrification of all buildings in the U.S. could lead to a one-third increase in annual electricity consumption. Another modeling effort[19] found that space heating electrification in Texas can reduce residential building energy use by 0–11%, although the magnitude depends on the efficiency assumed for heat pumps. These studies suggest the wide range of possible building energy use changes resulting from a variety of future building efficiency assumptions. Similarly, building energy saving also varies (e.g., by 4–30%) with different assumptions regarding occupant behavioral changes[42,43]. Despite the important role that income plays in determining building energy use in low- and middle-income countries[17,44], several studies have suggested very low household income elasticity of residential building energy use in the U.S. when other building characteristics are controlled[45,46].

Nevertheless, here we hold the characteristics of the building stock in each urban area fixed[3,5,47] at the reference-decade levels. This is mainly because the current building stock and its occupant behavior are well understood from extensive survey data. By focusing on the impacts of climate change, population growth, and power sector decarbonization, our findings establish a baseline for future research to delve into the complex socio-economic impacts on city-scale building energy use under each future scenario. The explicit representation of these impacts is possible considering the physics-based nature of the proposed bottom-up approach, which can incorporate projected changes in building shells, equipment efficiency, and occupant behavior. Furthermore, our high-granularity approach opens up new possibilities for high temporal resolution analyses on the intricate interactions between building energy consumption, building heat emission, and electricity generation that are sensitive to weather and climate changes.

## Methods

### A bottom-up approach for quantifying urban building energy use

To evaluate the behaviors and responses of city-scale building energy consumption to climate change in the U.S., we developed a bottom-up

approach that integrates physics-based building energy models and statistical methods (Supplementary Fig. 25). At the core of this approach lies the modeling of representative buildings, or prototype buildings, in each urban area, which relies on a whole building energy simulation program (EnergyPlus) driven by hourly weather data (historical reference period: 2010–2019 or the 2010s; future period: 2050–2059 or the 2050s). However, prototype buildings are inadequate in describing heterogeneous building characteristics and end-users' behaviors across the entire building stock. To address this limitation, we used the high-granularity End-Use Load Profiles (EULP) database[22] to scale up the simulated site energy use (hereafter simply energy use, unless otherwise specified) from individual building level to building stock level in each urban area. We then trained location-specific statistical models based on historical data from the EULP dataset to calibrate our city-scale results. These statistics-based calibration models aim to minimize discrepancies between the scaled results and the EULP dataset while capturing usage patterns that are potentially missing in the prototype models.

### Selection of U.S. urban areas

Urban areas in the CONUS in this study are primarily based on the urbanized areas identified by the U.S. Census Bureau, each with a population of at least 50,000 people. In comparison, the most granular level of the EULP dataset used for scaling and calibration is the Public Use Microdata Area (PUMA) level[22]. PUMAs are the smallest geographic units used by the U.S. Census Bureau for the tabulation and dissemination of the American Community Survey (ACS) Public Use Microdata Sample data. Each PUMA contains at least 100,000 people (and usually fewer than 200,000 people) and is much smaller in urban areas than in rural areas. To reconcile the mismatch between urban areas and PUMAs, we identified one or more PUMAs to represent each urban area. More specifically, a PUMA is selected if at least 10% of its spatial extent is within the boundaries of an urban area (hereafter representative PUMAs). On the one hand, this threshold ensures that we cover a sufficiently large number of CONUS urban areas. On the other hand, the selection also covers part of the urban periphery which may undergo future urban development and potential population growth. In cases where a PUMA spans multiple urban areas, we assigned it to the urban area that encompasses the largest proportion of the PUMA's spatial extent to avoid double counting. Following these procedures, we excluded urban areas without any representative PUMAs and retained 277 out of 481 CONUS urban areas in this study. These 277 urban areas cover 1,540 PUMAs across the country (Supplementary Figs. 26, 27). Approximately 44% of urban areas can be represented by a single PUMA, while the largest urban area, New York-Newark in NY-NJ-CT, is represented by 144 PUMAs (Supplementary Data 1).

### Historical hourly weather data

Physics-based building energy consumption modeling in each urban area requires hourly meteorological forcing data from representative weather stations. The National Renewable Energy Laboratory (NREL) identified 936 weather stations over the CONUS in the Typical Meteorological Year (TMY3) dataset and classified these stations based on their data uncertainty and completeness of the long-term historical observations; Class I and Class II stations have lower uncertainty data and more complete period of record than Class III stations[48]. In addition, a previous study[49] has demonstrated that using a subset of 216 TMY3 stations is sufficient to capture the granularity of different climate conditions in the CONUS. Considering the data quality, distance to urban areas, elevation, and climate types, we further improved the granularity by selecting 252 urban weather stations to represent 277 CONUS urban areas (Supplementary Fig. 27). More than 97% of the selected stations are either Class I or Class II stations. This selection results in 303 unique pairs of urban areas and weather stations, with 17

urban areas having more than one representative station (Supplementary Data 1).

We then compiled hourly meteorological data for the selected stations covering a 22-year historical period from 1998 to 2019. We retrieved quality-controlled hourly observations of air temperature, dew point temperature, sea level pressure, wind direction, wind speed, and accumulated liquid precipitation from the Integrated Surface Database (ISD)[50]. Post-processing was performed to correct mismatches in weather station identifiers and geographical locations. Solar radiation-related variables, including global horizontal radiation, direct normal radiation, diffuse horizontal radiation, and sky cover, were retrieved from the half-hourly, 4-km National Solar Radiation Database (NSRDB)[51]. These variables were computed using the Physical Solar Model (PSM)[52], which is a two-step physical model driven by cloud properties, atmospheric profiles, aerosol properties, and albedo from multiple datasets, and are available since 1998. All raw data were converted from Coordinated Universal Time (UTC) to local time. Sea level pressure was converted to surface pressure at each station using the hypsometric equation. A multi-step gap-filling approach was applied to fill gaps in hourly observations based on data from nearby stations and reanalysis data from the Modern-Era Retrospective Analysis for Research and Applications version 2 (MERRA-2)[53] (see further details in Supplementary Method 1).

### Future hourly weather data

Future hourly weather data at the locations of representative urban weather stations were derived from the latest CMIP6 model outputs. Four emissions and concentration or warming scenarios, SSP1-2.6, SSP2-4.5, SSP3-7.0, and SSP5-8.5, were selected to represent a wide range of possible future socio-economic developments, land use, and emission trajectories with different global warming levels (Supplementary Method 2 and Supplementary Tables 2, 3). These four future scenarios have been prioritized as the Tier 1 scenarios in the CMIP6's Scenario Model Intercomparison Project (ScenarioMIP)[54] and are also part of the core set of illustrative scenarios in the Intergovernmental Panel on Climate Change (IPCC)'s Sixth Assessment Report. Compared with the commonly used combinations of SSP and Representative Concentration Pathways (RCPs) in the literature, these SSPX-Y scenarios enhance the alignment between the socio-economic backgrounds and the resulting emissions futures (Supplementary Method 2). Ten CMIP6 general circulation models and Earth system models were selected based on data availability, and each model has a variant with outputs available for both historical simulations (hindcasts) and future projections under all four SSPX-Y scenarios (Supplementary Table 4). This ensures a consistent set of model variants across different scenarios.

We retrieved primarily 3-hourly climate model outputs of near-surface air temperature, near-surface specific humidity, surface pressure, eastward near-surface wind, northward near-surface wind, surface pressure, surface downwelling shortwave radiation, and surface diffuse downwelling shortwave radiation (Supplementary Method 3). Additionally, we retrieved daily maximum and minimum near-surface air temperatures to capture daily extremes (Supplementary Method 3). Raw model outputs were converted from UTC to local time and interpolated from model grids to station locations using inverse distance weighting. Linear interpolation was used to downscale temperature, humidity, wind components, and pressure to hourly data, while hourly solar elevation angle was used to downscale radiation data[55]. Relative humidity was derived from air temperature, specific humidity, and surface pressure. Direct normal radiation was calculated using shortwave radiation, diffuse shortwave radiation, and solar zenith angle.

We then applied the quantile delta mapping (QDM) method[56] to correct biases in the downscaled hourly climate model outputs. This method has proven effective in correcting systematic biases while

explicitly preserving the projected changes in each quantile[56]. Non-parametric cumulative distribution functions (CDFs) approximated using kernel density estimates were further adopted to improve the performance of the original QDM method[57], and a 3-month moving window was used to fit all CDFs. Following previous studies[56,58], the additive form of QDM was used for temperatures and pressure, whereas the multiplicative form was used for wind, humidity, and radiation variables. Similar to the treatment of historical weather data, quality control was carried out to smooth out outliers and ensure that variables satisfy psychrometric relationships.

## Physics-based building energy modeling

We used a state-of-the-art whole building energy simulation model, EnergyPlus, to produce building-level energy consumption data under historical and future climate conditions. EnergyPlus simulates the building and its energy systems based on heat balance principles, and encompasses various numerical modules capable of solving not only thermal zone conditions and the dynamics of heating, ventilation, and air conditioning (HVAC) systems, but also more complex behaviors such as interzone air mixing and fenestration systems. For each representative urban weather station, the response of building energy use to climate change was evaluated using a set of residential and commercial prototype building models developed by the Pacific Northwest National Laboratory (see details in Supplementary Method 4). These prototype buildings cover different building types, heating system types, and foundation types and have been widely used for cost-effectiveness analysis of building codes and climate impact assessments across the U.S. (e.g., refs. 59–61). Specifically, we used 40 residential prototype buildings (2 building types × 4 foundation types × 5 heating system types) and 28 commercial prototype buildings (14 building types × 2 heating system types) at each station (Supplementary Method 4). Because prototype building files are only available at representative locations, we further updated site information, design conditions, and water mains temperature to reflect city-specific characteristics. This generated a set of 20,604 prototype building models (68 archetypes × 303 station–city pairs).

Historical simulations span 10 years from 2010 to 2019 (the reference decade) to avoid the potential impact of the COVID-19 pandemic on energy use. We performed a total of 206,040 simulations (20,604 archetypes × 10 years) for this decade using EnergyPlus. Future simulations also last for a decade (2050–2059), but a decadal run was carried out for each climate model under each SSPX-Y scenario, resulting in a total of ~8.24 million simulations (20,604 archetypes × 10 years × 10 climate models × 4 SSPX-Y scenarios). All EnergyPlus runs were performed in parallel mode on the Sherlock cluster at Stanford University. Each simulation was conducted using 15-min timesteps for zone heat balance model calculation, and electricity, natural gas, fuel oil, and propane consumption data for different end-use categories were output at an hourly resolution.

## Fine-granularity end-use energy consumption data

Another cornerstone of the bottom-up approach for urban building energy use is a fine-granularity end-use (site) energy consumption dataset, the EULP database[22], primarily developed by the NREL. Harnessing numerous data sources and the two best-in-class building stock energy models of the U.S. residential and commercial sectors (ResStock and ComStock), the EULP dataset provides 15-min resolution load profiles for all major residential and commercial building types and end uses as well as building stock characteristics for all locations in the CONUS, and is one of the most comprehensive building load databases in the U.S. In particular, the two building stock models are notable for their ability to achieve high temporal and spatial resolutions. Using statistical distributions of building characteristics, they simulate ~900,000 individual building models to represent the energy consumption patterns of the entire U.S. building stock.

Some major data sources used for developing building stock models include building codes, the U.S. Energy Information Administration's Residential Energy Consumption Survey (RECS) and Commercial Buildings Energy Consumption Survey (CBECS), and the U.S. Census Bureau's American Housing Survey (AHS) and ACS. For both residential and commercial building stocks, the sensitivity of model parameters was first quantified with trained random forest models. These input parameters were then adjusted following a region-by-region calibration approach to improve the consistency between the results and additional empirical data (e.g., hourly advanced metering infrastructure data and energy sales data). Extensive and thorough validation across different spatial and temporal scales has been carried out using public and private datasets, demonstrating the accuracy of this dataset in characterizing the U.S. building stock. Further details of the two building stock models and the calibration and validation of the EULP database are provided in Supplementary Method 5.

It is noteworthy that the physics-based simulations in the EULP database also rely on EnergyPlus driven by hourly weather data. Given this inherent similarity, we anticipated that the response of individual loads (especially weather-dependent ones, i.e., heating and cooling) to historical weather simulated in our study would align with the EULP data. In addition, to use the EULP database in our city-scale calibration models, we aggregated the 15-min, PUMA-level EULP outputs of residential and commercial building energy consumption to the hourly scale for each of the 277 U.S. urban areas (Supplementary Table 5). PUMA-level building stock characteristics were also aggregated to the city scale. For each urban area, the simulated hourly energy consumption was then scaled and aggregated based on the shares of building types, foundation types, and primary heating types in the city-scale building stock, similar to previous national and city-scale studies[60,61].

## Statistical calibration models

As expected, due to the similarity in simulation algorithm and meteorological forcing, the patterns of city-scale EnergyPlus-based hourly building energy consumption for individual end-use categories (e.g., heating or cooling) are generally consistent with those in the EULP data (see diurnal and monthly comparisons of example urban areas in Supplementary Figs. 28, 29). However, the sum of these individual end-use categories often exhibits large discrepancies in both magnitude and load shapes, even on the monthly scale (Supplementary Fig. 30). This is mainly caused by the lack of diverse building characteristics, operations, and controls in our prototype-based simulations.

To reduce these discrepancies and better capture the response of diverse city-scale building stock to weather conditions, we implemented a post hoc calibration approach using a set of location-specific statistical models. For each end-use load category $i$ at time $t$ in an urban area from the EULP database, $E_{i,t}$, following a commonly used dose–response function form[5,17,47], we trained a statistical model using a yearlong hourly time series (training datasets; see Supplementary Method 5),

$$E_{i,t} = \alpha_i E_{i,t}^a + \sum_j \beta_{j,i} T_{j,t} + \omega_h + \psi_m + \phi_w + \sum_k \gamma_{k,i} \eta_{k,t} E_{i,t}^a + C_t + \epsilon_{i,t} \quad (1)$$

where $E_{i,t}^a$ is the aggregated result of EnergyPlus-based end-use load category $i$ at time $t$; $\omega_h$, $\psi_m$, and $\phi_w$ are hour-of-day, month-of-year, and weekend fixed effects, respectively; $\eta_{k,t} E_{i,t}^a$ denotes the interactions between the aggregated energy use and hour-of-day dummy variables, as a correction of diurnal load shape; $C_t$ denotes six-order Chebyshev polynomials, reflecting the impacts of low-frequency factors[62]; $\epsilon_{i,t}$ is the error term; and $\alpha_i$, $\beta_{j,i}$, and $\gamma_{k,i}$ are regression coefficients. $T_{j,i}$ is a dummy variable for the hourly air temperature at time $t$ that falls into a

temperature bin $j$, representing the potential nonlinear response of the relationships to ambient temperature[47]. Different from previous studies that used an equal width for all temperature bins (e.g., 3 °C in ref. [5]), we used the 2nd, 5th, 95th, and 98th percentiles as well as nine deciles of the distribution as the bins' cutoffs to ensure that each bin has sufficient data samples.

We then post-calibrated all weather-dependent end-use load categories (i.e., electricity use for cooling, electricity use for space heating, natural gas use for space heating, fuel oil use for spacing heating, and propane use for space heating) with the trained calibration models separately for residential and commercial buildings in each urban area. Additionally, we further trained calibration models for water heating energy consumption in residential buildings due to the stochastic occupant behavior model used in ResStock[22]. Then for each urban area, the hourly time series of residential and commercial building energy consumption, $E_t$, can be computed as

$$E_t = \sum_i \hat{E}_{i,t} + \sum_k E_{k,t} \qquad (2)$$

where $\hat{E}_{i,t}$ is the calibrated end-use load estimated using Eq. (1), and $E_{k,t}$ denotes other less- or non-weather-dependent loads from the validated EULP database, which are nearly identical in different years. Equations (1) and (2) were used to derive the time series of both historical ($E_{i,t,\text{his}}$ or $E_{t,\text{his}}$) and future ($E_{i,t,\text{fut}}$ or $E_{t,\text{fut}}$) building energy uses. Note that the future site energy consumption in Eq. (2) does not involve any impacts from future population change.

To examine whether the calibration models can be used under different weather conditions and climate change scenarios, we applied the trained models to generate new yearlong time series of all variables and compared the results with two independent testing datasets from the EULP (Supplementary Method 5). The estimated city-scale residential and commercial building end-use loads demonstrate great agreement with the EULP data across different temporal scales (from hourly to annual) for both training and testing periods (Supplementary Figs. 28–32 and Supplementary Tables 6–8). In particular, the Pearson's $r$ values for the comparisons between the calibrated total building energy use and the EULP data are well above 0.99 (Supplementary Figs. 31, 32).

### Response of urban building stock to future population changes

Spatially and temporally varied future urban population changes will have an impact on the building stock and the associated building energy consumption. For future population change, we use a 1-km, decadal population projection dataset, which was downscaled from global 1/8° projections[63] and is quantitatively and qualitatively consistent with SSPX-Y scenarios (Supplementary Method 2). The gridded population projections were aggregated to the city scale, and population counts for the 2010s and 2050s were derived from the average of 2010 and 2020 data and the average of 2050 and 2060 data, respectively. Considering that the base year of these projections is 2000, we used the population counts from the EULP and the ratio of the 2050s population to the 2010s population to derive the corrected population projections under each of the four SSPX-Y scenarios (Supplementary Fig. 23).

Early studies have demonstrated that urban building energy consumption or its proxy (e.g., nightlights) nearly linearly scale with urban population[64,65], i.e., the exponent $\delta$ in the following power-law relationship between population $P$ and an urban property $Y$ (e.g., building energy use) at time $t$ roughly equals 1.0,

$$Y_t = \lambda P_t^{\delta} \qquad (3)$$

where $\lambda$ is a normalization constant. Similar relationships were observed using the EULP database for the selected 277 urban areas in

the CONUS: $\delta = 1.013$ ($R^2 = 0.926$) for $P$ = population, and $\delta = 1.012$ ($R^2 = 0.948$) if the base in Eq. (3) is the total floor area. This suggests that building energy consumption linearly scales with both population and floor area in these urban areas (Supplementary Fig. 33). Therefore, we assumed that the future building stock and building energy consumption change proportionally with the population in the selected urban areas[3,59,66]. Mathematically, the time series of future site energy consumption in an urban area, $E_{\text{SI},t,\text{fut}}$, can be calculated as

$$E_{\text{SI},t,\text{fut}} = E_{t,\text{fut}} \frac{P_{\text{fut}}}{P_{\text{his}}} = E_{t,\text{fut}} \frac{F_{\text{fut}}}{F_{\text{his}}} \qquad (4)$$

where $E_{t,\text{fut}}$ is the time series of energy consumption with building stock and population fixed at historical levels but under future climate conditions computed using Eq. (2), and F is the total floor area, with the subscripts "his" and "fut" denoting historical and future periods, respectively. The subscript "SI" represents site energy consumption that includes impacts from population change.

The above relationship has important implications for changes in energy use intensity (EUI), an indicator used in this study to isolate the impact of climate change from that of population change. The relative change in EUI under future conditions, when compared with historical conditions, can be calculated as $(E_{\text{SI},t,\text{fut}}/F_{\text{fut}} - E_{\text{SI},t,\text{his}}/F_{\text{his}})/(E_{\text{SI},t,\text{his}}/F_{\text{his}})$. Substituting Eq. (4) into this expression yields

$$(E_{\text{SI},t,\text{fut}}/F_{\text{fut}} - E_{\text{SI},t,\text{his}}/F_{\text{his}})/(E_{\text{SI},t,\text{his}}/F_{\text{his}}) = (E_{t,\text{fut}} - E_{t,\text{his}})/E_{t,\text{his}} \qquad (5)$$

where $E_{\text{SI},t,\text{his}} = E_{t,\text{his}}$, as the historical floor area is constant. Equation (5) suggests that the impact of climate change on EUI, when isolated from the impact of population change, is equivalent to imposing future climate conditions onto the current building stock.

### Response of urban air conditioning saturation to future warming

Future warming will also increase the market penetration of air conditioners, especially in regions where the current penetration rate is relatively low, such as the Northwest (on average ~52% for residential buildings based on the EULP database). Previous studies[8,67] used an empirical model to determine future saturation based on the current rate and cooling degree days. However, the empirical model was trained using data from only 39 U.S. urban areas. Here we used historical residential building data from 277 CONUS urban areas and developed an exponential saturation function,

$$S_0 = 0.948 - 0.824 e^{-0.00343 CDD_0} \qquad (6)$$

where $S_0$ is the AC saturation rate in the 2010s, and $CDD_0$ is the mean annual cooling degree days during the same period. We then used the slope of this saturation curve to project the future saturation rate in the 2050s for each urban area,

$$S_t = S_0 + 0.00282 e^{-0.00343 CDD_t}(CDD_t - CDD_0) \qquad (7)$$

where $S_t$ is the future AC saturation rate during 2050–2059, and $CDD_t$ is the mean annual cooling degree days in the 2050s. We derived $S_t$ separately for each climate model projection under each SSPX-Y scenario. This ensures the diverse response of market penetration in future CONUS urban areas under different climate projections.

For commercial buildings, a linear saturation curve was fitted based on historical data from the EULP. However, the regression function has a very low $R^2$ (0.112), and the change of AC saturation rate induced by warming estimated using the curve is one order of magnitude smaller than its residential counterpart. In addition, the current AC penetration rate of commercial buildings in the selected 277 urban

areas is in general much higher than that in residential buildings. For commercial buildings, on average 98.9% of 277 urban areas have a saturation rate higher than 80% (cf. 73.6% of urban areas for residential buildings). Therefore, we only considered saturation rate change for the residential sector.

## Linear regressions for the response of space-conditioning energy use intensity to warming

Previous studies have identified statistically significant linear or non-linear relationships between the relative change in electricity use and end-of-century global warming extent[3,25]. However, we did not observe such relationships in this study, which mainly results from the spatially varied compensation between increasing cooling demand and decreasing space heating demand under mid-century warming conditions. Nevertheless, we observed significant linear dependence of both building cooling demand and space heating demand on warming. For each urban area, the projected changes in annual cooling demand, space heating demand, and air temperature from a wide spectrum of 40 potential trajectories (10 CMIP6 climate models × 4 SSPX-Y scenarios) provide a sufficiently large sample size to build linear regression models (Supplementary Fig. 34). Of particular interest is the slope of the linear regression model, which indicates the city-scale response of the cooling or space-heating EUI to a local temperature change of 1 °C (for relative change the unit is %/°C).

We fitted linear regression models between the city-scale air temperature change and mid-century changes in E-EUI for cooling, E-EUI for space heating, natural gas EUI for space heating, fuel oil EUI for space heating, and propane EUI for space heating with the ordinary least squares method. The linear regression models for these five space-conditioning EUIs are statistically significant for all urban areas ($p$ value $< 10^{-8}$). The minimum $R^2$ values for the responses of E-EUI for cooling, E-EUI for space heating, natural gas EUI for space heating, fuel oil EUI for space heating, and propane EUI for space heating are 0.757, 0.666, 0.673, 0.716, and 0.596, respectively. In addition, the warming-response of E-EUI for space cooling with AC saturation rate fixed at the reference-decade level exhibits a spatial pattern similar to the case with changing AC saturation rate (Fig. 4a and Supplementary Fig. 17a), although with a relatively smaller magnitude of response. This suggests the robustness of responses quantified using linear regression models (e.g., Supplementary Fig. 34).

## Conversion to source energy use

It is noteworthy that the end-use energy consumption of buildings (site energy consumption) does not account for energy losses during generation, transmission, and distribution. To assess the actual primary energy used by buildings (source energy consumption), national-level site-to-source conversion factors have been commonly used in existing impact assessments (e.g., ref. 59). However, the use of uniform conversion factors across the entire country can largely overlook the substantial regional variability of fuels and resources used in electricity generation[21].

To convert site electricity use to source energy consumption (Supplementary Fig. 25), we derived spatially varied site-to-source conversion factors based on the electricity generation data from the Cambium dataset[68] (see Supplementary Method 6 for further details). The Cambium dataset offers biennial projections of structural evolvements in the U.S. electric power sector through 2050 under different scenarios using outputs from the Regional Energy Deployment System (ReEDS) model, the Distributed Generation Market Demand (dGen) model, and the commercial production cost model (PLEXOS). For the evolution of the U.S. electric power sector, we consider two scenarios: the business-as-usual scenario and the zero-carbon scenario.

For the business-as-usual scenario, we used outputs from the Mid-case projection, which prescribes mid-level demand growth, system cost, resources, prices, and technology inputs. This business-as-usual

scenario represents the current views of energy technologies. It only reflects the impacts of existing state, regional, and federal carbon policies enacted as of June 2021, with no new carbon policies applied in the future. The zero-carbon scenario is a variant of the Mid-case projection, which assumes a drastic decarbonization of the U.S. electric power sector. Under this scenario, $CO_2$ emissions from the electric power sector plummet to 95% below 2005 levels by 2035 and eventually achieve net zero (100% reduction) by 2050. This zero-carbon scenario is close to the carbon-pollution-free electricity goal set in the U.S. long-term strategy. The evolution of the generation mix under these two scenarios is shown in Supplementary Fig. 18. It is noteworthy that the two power sector scenarios employed in this study focus solely on the possible changes in the composition of electricity generation at the regional scale (the U.S.) rather than at the global scale. The combination between power sector scenarios and SSPX-Y scenarios enables us to probe the sensitivity of building source energy consumption in U.S. urban areas to global climate change, in which the regional power sector scenarios are not informed by the global SSPX-Y scenarios[68].

The finest geographic unit of the generation mix data from the Cambium dataset is the balancing area (BA) (Supplementary Fig. 35), which acts as the basic node to balance supply and demand. Nevertheless, conversion factors estimated from the BA-level generation mix overlook the impact of imported and exported energy between balancing areas. To minimize the mismatch between electricity generation and consumption, we aggregated these BAs into 20 generation and emission assessment (GEA) regions. These GEA regions closely resemble the U.S. Environmental Protection Agency (EPA)'s eGRID subregions defined to limit electricity imports and exports. We adopted changing heat rates of different primary energy types under business-as-usual and zero-carbon scenarios to derive source energy consumption (Supplementary Table 9). For non-combustible renewable energy resources, we calculated the equivalent primary energy consumption following the EPA's fossil fuel equivalency approach (Supplementary Method 6). For other site fossil fuel consumption (natural gas, fuel oil, and propane), we applied a set of national-level conversion factors (Supplementary Method 7).

## Decomposing the drivers of changes in building source energy use

Source energy consumption of urban buildings under different future scenarios is collectively influenced by climate change, population change, and changes in energy sources and technologies used for electricity generation (generation mix). The total source energy consumption of buildings in an urban area, $E_{SO}$, can be computed as

$$E_{SO} = \sum_i EUI_{SI,i} \times F \times CF_i \tag{8}$$

where $i$ denote site energy type (electricity, natural gas, fuel oil, or propane), and $EUI_{SI}$, $F$, and $CF$ are site EUI, total floor area, and the site-to-source conversion factor, respectively. Here site EUI for energy type $i$ equals total site energy use for this energy type, $E_{SI,i}$, divided by the floor area of the entire building stock. Then the above equation can be rewritten as

$$E_{SO} = \sum_i \frac{E_{SI,i}}{F} \times F \times CF_i \tag{9}$$

After substituting Eq. (4) into Eq. (9), the total source energy use for the future period can be written as,

$$E_{SO,fut} = \sum_i E_{i,fut} \times \frac{F_{fut}}{F_{his}} \times CF_{i,fut} = \sum_i E_{i,fut} \times \frac{P_{fut}}{P_{his}} \times CF_{i,fut} \tag{10}$$

where $E_\text{fut} = \sum_t E_{t,\text{fut}}$, the annual site energy consumption of current building stock under future climate conditions. Renaming the population ratio as $PR$ yields

$$E_\text{SO,fut} = \sum_i E_{i,\text{fut}} \times PR_\text{fut} \times CF_{i,\text{fut}} \tag{11}$$

Similarly, for the historical period we have

$$E_\text{SO,his} = \sum_i E_{i,\text{his}} \times PR_\text{his} \times CF_{i,\text{his}} \tag{12}$$

where $PR_\text{his} = 1$, as the population is constant for the historical period. Generalizing Eqs. (11) and (12) leads to

$$E_\text{SO} = \sum_i E_i \times PR \times CF_i \tag{13}$$

with the three terms on the right-hand side involving the impacts of climate change, population dynamics, and changes in the electric power sector, respectively. In particular, the climate change component here is consistent with Eq. (5).

We then applied the logarithmic mean Divisia index (LMDI) method[28] to decompose the contributions of these three drivers to total source energy consumption changes. Using an additive decomposition form, the change in total source energy consumption of buildings in an urban area can be attributed as

$$\Delta E_\text{SO} = \Delta E_E + \Delta E_{PR} + \Delta E_{CF} \tag{14}$$

where

$$\Delta E_E = \sum_i \frac{E_{\text{SO},i,\text{fut}} - E_{\text{SO},i,\text{his}}}{\ln E_{\text{SO},i,\text{fut}} - \ln E_{\text{SO},i,\text{his}}} \ln\left(\frac{E_{i,\text{fut}}}{E_{i,\text{his}}}\right) \tag{15}$$

$$\Delta E_{PR} = \sum_i \frac{E_{\text{SO},i,\text{fut}} - E_{\text{SO},i,\text{his}}}{\ln E_{\text{SO},i,\text{fut}} - \ln E_{\text{SO},i,\text{his}}} \ln\left(\frac{PR_{i,\text{fut}}}{PR_{i,\text{his}}}\right) \tag{16}$$

$$\Delta E_{CF} = \sum_i \frac{E_{\text{SO},i,\text{fut}} - E_{\text{SO},i,\text{his}}}{\ln E_{\text{SO},i,\text{fut}} - \ln E_{\text{SO},i,\text{his}}} \ln\left(\frac{CF_{i,\text{fut}}}{CF_{i,\text{his}}}\right) \tag{17}$$

We further conducted an uncertainty analysis using a modified elasticity-based decomposition method[69] (see Supplementary Method 8). Unlike the LMDI method which provides perfect decomposition, the elasticity-based decomposition leaves an unexplained residual term after allocating the contributions of drivers. For the attribution of changes in future source energy use in this study, the LMDI method and the elasticity-based method yield very similar results (Supplementary Fig. 36), suggesting the reliability and robustness of the estimated contributions.

### Reporting summary
Further information on research design is available in the Nature Portfolio Reporting Summary linked to this article.

## Data availability
The boundaries of urban areas and Public Use Microdata Areas (PUMAs) are extracted from the U.S. Census Bureau's TIGER/Line Shapefiles, which are available at https://www.census.gov/geographies/mapping-files/time-series/geo/tiger-line-file.html. Station-based hourly weather observations are from the Integrated Surface Database (ISD) developed by the National Centers for Environmental Information, which is available at https://www.ncei.noaa. gov/products/land-based-station/integrated-surface-database. Historical typical meteorological year data, also derived from the ISD, are available at https://energyplus.net/weather or https://climate. onebuilding.org/sources/default.html. Historical radiation data and MERRA-2 reanalysis data are available from the National Solar Radiation Database (NSRDB) at https://nsrdb.nrel.gov/. The Coupled Model Intercomparison Project Phase 6 (CMIP6) climate projections used in this study are available through the CMIP6 Search Interface at https://esgf-node.llnl.gov/search/cmip6/. The complete dataset of the End-Use Load Profiles (EULP) for residential and commercial buildings is available at https://data.openei.org/submissions/4520. The Standard Scenarios and Cambium datasets are available at https://scenarioviewer.nrel.gov/. Downscaled high-resolution population projections are available at https://sedac.ciesin.columbia.edu/data/set/popdynamics-1-km-downscaled-pop-base-year-projection-ssp-2000-2100-rev01. Source data are provided with this paper.

## Code availability
The whole building energy simulation program used in this study, EnergyPlus version 9.0.1, is available at https://github.com/NREL/EnergyPlus/releases/tag/v9.0.1. Residential and commercial prototype building models are available at https://www.energycodes.gov/prototype-building-models. ComStock and ResStock are available at https://github.com/NREL/resstock and https://github.com/NREL/ComStock, respectively. Algorithms for data processing, analysis, and visualization are based on MATLAB R2020b, R 4.1.2, and R 4.2.2 and are available from the corresponding author upon request.

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

## Acknowledgements

We acknowledge the high-performance computing support from the Sherlock cluster provided by Stanford University and the Stanford Research Computing Center. We would like to thank Carlo Bianchi (National Renewable Energy Laboratory, USA), Linda Lawrie (Climate.OneBuilding.Org), and Dru Crawley (Climate.OneBuilding.Org) for their assistance with historical weather data processing. We would like to thank Seiji Yukimoto (Meteorological Research Institute, Japan), Lijuan Li (Institute of Atmospheric Physics, Chinese Academy of Sciences, China), Swapna Panickal (Indian Institute of Tropical Meteorology, India), Young-Hwa Byun (National Institute of Meteorological Sciences, South Korea), Jae-Hee Lee (National Institute of Meteorological Sciences, South Korea), and Gavin A. Schmidt (National Aeronautics and Space Administration, USA) for their help with the processing of future climate projections. We are grateful to Pieter Gagnon (National Renewable Energy Laboratory, USA) for his assistance with the future projections of the U.S. electric power sector. We would also like to express our appreciation to Anders Ahlström (Lund University, Sweden) for his comments in the early stage of this study. Financial support for publication was partially provided by the University of Oklahoma Libraries' Open Access Fund. S.F. acknowledges support from ANID (ANILLO ACT210046) and CORFO (19BP-117358). This work was authored in part by the National Renewable Energy Laboratory, operated by Alliance for Sustainable Energy, LLC, for the U.S. Department of Energy (DOE) under Contract No. DE-AC36-08GO28308. Funding was provided by U.S. Department of Energy Office of Energy Efficiency and Renewable Energy Building Technologies. The views expressed in the article do not necessarily represent the views of the DOE or the U.S. Government. The U.S. Government retains and the publisher, by accepting the article for publication, acknowledges that the U.S. Government retains a non-exclusive, paid-up, irrevocable, worldwide license to publish or reproduce the published form of this work, or allow others to do so, for U.S. Government purposes.

## Author contributions

C.W., J.S., and R.B.J. conceived the study. C.W., S.F., and Y.L. analyzed historical and future weather data. C.W. developed the modeling procedure and conducted building energy simulations with assistance from D.S. C.W., J.L.R., and H.H. compiled the city-scale end-use load profiles and building stock information. C.W., Y.Z., and Z.O. carried out the population change analysis. C.W. performed the site-to-source conversion analysis. C.W. wrote the first draft of the manuscript. All authors reviewed and contributed to the manuscript.

## Competing interests

The authors declare no competing interests.
