## [Peer Review File · Nature Communications]

Impacts of climate change, population growth, and power sector decarbonization on urban building energy useEditorial Note: Parts of this Peer Review File have been redacted as indicated to remove third-party material where no permission to publish could be obtained.

REVIEWER COMMENTS

Reviewer #1 (Remarks to the Author):

Review of "Impacts of climate change, population growth, and power sector decarbonization on building energy use in U.S. cities"

This manuscript estimates the mid-century impact on residential and commercial building energy use in 277 individual cities in the US from a combination of climate change (multiple standard scenarios), population change, and energy supply decarbonization. The authors should be applauded for the herculean effort undertaken here in combining a large and diverse array of data, models and analysis. The methods offer novel solutions to problems of building energy estimation at fine space and time scales.

While the numerical analysis in the paper appears sound and thorough, the paper suffers from a few basic communication problems. The first is grammar and phrasing. This needs considerable work. As a courtesy to the authors and in compensation for the delay in getting my review completed, I have performed editing with track changes in the manuscript to assist the authors in this task.

In my track changes I have also included a series of comments that are at times of a substantive/conceptual nature. I will highlight the main themes of those here (there are many additional/smaller comments in the track changed manuscript – many about clarity and confusion of terms and phrases) but note that detailed versions of what I write here are within the track changed manuscript as well, particularly in specific places where the questions/comments arise.

First the manuscript needs a better introductory setup, reviewing the many factors that will alter future building energy demand. The authors are tackling three in this manuscript (climate change, population change, and decarbonization) but the reader should be made aware of other critical drivers of future building energy use and these can probably be categorized broadly as "building technology", "behavioral change", and "economics". Then the reader has better context for the subset of drivers this paper explores. This context emerges at various points in the manuscript (or not at all) and that presentation is confusing and at times inconsistent with the flow of the paper. I consider it critical to give the reader this more comprehensive understanding so that this work, and the elements it takes on can be understood within the wider context.

The nature of both the EUI changes and the total source energy changes were often hard to understand in terms of what sub-elements of EUI/energy were being discussed. At times it was not clear of the total EUI was just the sum of the cooling/heating sub-elements or reflected the complete building EUI which would include the many other energy demands in buildings beyond heating/cooling (appliances, lighting and so forth). Better clarity on that is important. I made recommendations about using a somewhat different set of phrases (and doing so throughout the manuscript) to help with this. At the least, good definitions and then consistent use is essential.

Though the manuscript spends a lot of text on the impact of climate change, the reader discovers towards the end that indeed population and decarbonization are the primary drivers and climate change is a distant third (other than the implications for extreme events which are an important climate change impact and deserving of emphasis). That is fine but given the relative importance, the details of population and decarbonization appear much less explored or described. For example, 4 of the 6 figures in the main manuscript present the impacts of climate change, the 5th brings in the decarbonization and the 6th reflects population change (in addition to the other two). The same proportions exist in the supplementary figures. Again, there is nothing wrong with that, but given the relative contributions in the final outcomes it seems more detail methodologically and in the analysis should be afforded to the population and decarbonization components.

The impact of decarbonization remains quite unclear to me. The explanation for decarbonization found in the methods and Supp Mat. are difficult to follow. At first blush, it is not clear why shifting the supply mix or using direct capture, etc would have any impact on the electricity demand of a building – certainly the scope 2 emissions will change, but I don't see why the total energy demand would change. The only thing I can imagine is that the T&D losses may be less if the electricity supply is improved with decarbonization. Is that the reason? Or is some electrification of current on-site fossil fuel use involved? Furthermore, in looking at the results, decarbonization has a much larger impact than I would expect given what I know about T&D losses and the conversion factors used in this study. At the least, this needs additional explanation so that this important element is understood.

The discussion section, where the policy implications are discussed, is a bit difficult to follow in places (paragraph 3 in particular). The policy implication discussion is primarily speculative or a bit too cursory to really understand the implications. It is often stated that the spatial heterogeneity in the response will require a "diversity of approaches" to meeting energy demand. But, it isn't clear why that is truly needed and discussion of what those diverse approaches are is limited. The results indicate that population growth dominates the energy demand (the decarbonization may as well but as I have noted previously, I don't understand that element of the paper), hence, it seems that the bulk impact of population and the spatial distribution of the increases will be important to the heterogeneous impact. However, the population dataset used is fairly coarse (1 km but downscaled from a 1/8 degree projection.... Or about 16 km at midlats I believe). Hence, my sense is that just keeping up with that demand will need simple capacity expansion in both the electricity generation and fossil fuel (NG) delivery. I am not sure what this means for diversity of approaches as I did not see a spatial breakdown of the population impact other than the latitudinally binned output (figure 6) which shows very little spatial heterogeneity.

So, the wider impacts part of the paper – implications for energy supply capacity and resilience still seems a bit cursory or not as well thought out as the excellent work on building this methodology and getting these results. My impression looking at the results is that there will need to be significant capacity expansion of electricity generation everywhere with perhaps a bit more in the southern tier of cities. Since the grid is not generally "local" I don't understand what the diversity of approaches to power supply means in this context. Certainly resilience in time as extreme events increase, that is clear. But, the system requirements for the combined impact of population/decarbonization/climate change seems to demand basic bulk increase in capacity..... hopefully renewable forms.

My final recommendation is "accept with minor/major revision". I extend into major because it may take work to build more analysis on to the population and decarbonization aspects of the results and build a better discussion section with clearer analysis/discussion on the wider implications to energy supply. However, that may not be difficult for the authors.... that is difficult to assess.

Kevin Gurney
Northern Arizona University

Reviewer #2 (Remarks to the Author):

1. The terminology for the number of peak hours seems confusing. The peak hours normally refer to the time when HVAC demand reaches maximum. The increase of 170% is substantial and could be misleading. Recommend the authors present the absolute number of hours as well.
2. For each 1°C warming, city-level space-conditioning EUI on average rises and drops by 14% and ~10% for cooling and space-heating, respectively. The increase of 14% in cooling was only mentioned in the abstract. The analysis conducted in this study seems insufficient to draw this conclusion. Did the research team simulate and test the impacts of every 1°C interval within a temperature range on cooling and heating energy use?
3. What are the resources of the measurement data for statistical calibration?

4. How do EnergyPlus models integrated with the statistical model in this study? If EnergyPlus calculates the end-uses and is processed by the statistical model, the fitted data-driven model may not be extrapolated for future climate conditions since it is a data-driven model. This can result in poor quality of the simulation results, and thus the unfounded conclusion.

5. page 20, lines 451-455

“The sum of these individual end-use categories usually exhibits large discrepancies in both the magnitude and load shapes, even on the monthly scale. This is mainly induced by the lack of diverse building characteristics in our prototype-based simulations.”

This statement on discrepancies between prediction and measurement data is partially correct. The discrepancies between model prediction and measurements for buildings are because the prototype models do not represent the actual building design, operation, and control that is being simulated. The lack of diverse building characteristics in prototype-based simulation plays a less important role.

6. Hourly building energy consumption has been conducted for 277 U.S. cities. However, the analysis of simulation results seems limited. Recommend the author analyze the simulation results in-depth. What can we learn from these substantial modeling efforts?

Responses to Reviewer #1 (Dr. Gurney):

Review of “Impacts of climate change, population growth, and power sector decarbonization on building energy use in U.S. cities”

This manuscript estimates the mid-century impact on residential and commercial building energy use in 277 individual cities in the US from a combination of climate change (multiple standard scenarios), population change, and energy supply decarbonization. The authors should be applauded for the herculean effort undertaken here in combining a large and diverse array of data, models and analysis. The methods offer novel solutions to problems of building energy estimation at fine space and time scales.

While the numerical analysis in the paper appears sound and thorough, the paper suffers from a few basic communication problems. The first is grammar and phrasing. This needs considerable work. As a courtesy to the authors and in compensation for the delay in getting my review completed, I have performed editing with track changes in the manuscript to assist the authors in this task.

In my track changes I have also included a series of comments that are at times of a substantive/conceptual nature. I will highlight the main themes of those here (there are many additional/smaller comments in the track changed manuscript – many about clarity and confusion of terms and phrases) but note that detailed versions of what I write here are within the track changed manuscript as well, particularly in specific places where the questions/comments arise.

We appreciate the encouraging remarks and insightful and constructive comments from Dr. Gurney. We thoroughly revised the manuscript following your suggestions to improve clarity and enhance the policy implications. We also added new figures and sections to balance the description of climate change impacts and those of population growth and power sector decarbonization. Please see our point-by-point responses below. Note that the line numbers are based on the word document, not the PDF file converted by the submission system.

First the manuscript needs a better introductory setup, reviewing the many factors that will alter future building energy demand. The authors are tackling three in this manuscript (climate change, population change, and decarbonization) but the reader should be made aware of other critical drivers of future building energy use and these can probably be categorized broadly as “building technology”, “behavioral change”, and “economics”. Then the reader has better context for the subset of drivers this paper explores. This context emerges at various points in the manuscript (or not at all) and that presentation is confusing and at times inconsistent with the flow of the paper. I consider it critical to give the reader this more comprehensive understanding so that this work, and the elements it takes on can be understood within the wider context.

Thanks for the suggestion. In the revised Introduction section, we introduced that urban building energy use is influenced by meteorological conditions and climate change, socioeconomic developments such as population, household income, and electricity price, building technology

advancements, possible changes in individual behavior, and electric power sector evolutions. Please see Lines 37–64 in the revised manuscript. We also reviewed relevant studies on other drivers in the revised Discussion section to give our readers a more comprehensive understanding of this work. Please see Lines 370–385.

In the revised Introduction, we further highlighted that we only consider the impacts of climate change, population growth, and power sector decarbonization in this study (as suggested by the article title), while holding “building technologies, occupant behavior, and other socio-economic factors at their current levels, except for the AC penetration rate”, to “establish baseline projections for different future warming scenarios”. Please see Lines 74–76.

The nature of both the EUI changes and the total source energy changes were often hard to understand in terms of what sub-elements of EUI/energy were being discussed. At times it was not clear if the total EUI was just the sum of the cooling/heating sub-elements or reflected the complete building EUI which would include the many other energy demands in buildings beyond heating/cooling (appliances, lighting and so forth). Better clarity on that is important. I made recommendations about using a somewhat different set of phrases (and doing so throughout the manuscript) to help with this. At the least, good definitions and then consistent use is essential.

Thanks for pointing this out and for suggesting alternative phrases. To avoid potential further confusion, we did not use the suggested terms “metered energy” and “total energy”. Please see detailed explanations in our responses to comments on Line 12, Lines 49–53, and Line 68 below.

However, in the revised Introduction, we pointed out that site energy is “electricity and fossil fuels consumed in buildings” (Lines 58–59), while source energy is site energy plus “energy losses during generation, transmission, and distribution” (Line 59). In the revised Introduction section, we also added “site energy use of buildings (hereafter simply “energy use”, unless otherwise specified)” (Lines 77–78) and dropped “site” in the following sections to make the main text clearer and more concise.

Following the reviewer’s suggestions, we used electricity EUI (E-EUI) and non-electricity EUI (NE-EUI) in the revised manuscript. We mentioned that NE-EUI includes “natural gas, fuel oil, and propane EUIs” in Lines 81–82.

The term “total EUI” is the complete building EUI that includes all energy demands. In the revision, we clarified that (1) total E-EUI includes “space conditioning, water heating, and all other miscellaneous electric end-use types” (Lines 93–95); (2) the use of fossil fuels in buildings includes “space heating, water heating, and other miscellaneous end-use categories” (Lines 144–145); and (3) total EUI of buildings is “the sum of E-EUI and NE-EUI” (Lines 171–172).

For consistency, we added a new Supplementary Note 1 – “The use of ‘site’ in the captions of supplementary figures”. We clarified in the Supplementary Information that “we simplify ‘site energy use’ as ‘energy use’ in the captions of all supplementary figures, unless otherwise specified. This includes energy use intensity, which is calculated using site energy use.”

Though the manuscript spends a lot of text on the impact of climate change, the reader discovers towards the end that indeed population and decarbonization are the primary drivers and climate change is a distant third (other than the implications for extreme events which are an important climate change impact and deserving of emphasis). That is fine but given the relative importance, the details of population and decarbonization appear much less explored or described. For example, 4 of the 6 figures in the main manuscript present the impacts of climate change, the 5th brings in the decarbonization and the 6th reflects population change (in addition to the other two). The same proportions exist in the supplementary figures. Again, there is nothing wrong with that, but given the relative contributions in the final outcomes it seems more detail methodologically and in the analysis should be afforded to the population and decarbonization components.

Thanks for the suggestions. Following the suggestions, we added new figures to the revised manuscript and Supplementary. In the revised main text, 4 of the 8 figures now present results related to decarbonization and population change. We also expanded the Introduction, Discussion, and results sections to provide more details about power sector decarbonization and population dynamics. Please see Lines 57–62, Section “Impacts of future changes on total source energy use”, Section “Attributions of source energy use change”, Lines 307–322, Figs. 5–8, and Supplementary Figs. 19, 20, and 24. Please also see our responses to other related specific comments below.

The impact of decarbonization remains quite unclear to me. The explanation for decarbonization found in the methods and Supp Mat. are difficult to follow. At first blush, it is not clear why shifting the supply mix or using direct capture, etc would have any impact on the electricity demand of a building – certainly the scope 2 emissions will change, but I don't see why the total energy demand would change. The only thing I can imagine is that the T&D losses may be less if the electricity supply is improved with decarbonization. Is that the reason? Or is some electrification of current on-site fossil fuel use involved? Furthermore, in looking at the results, decarbonization has a much larger impact than I would expect given what I know about T&D losses and the conversion factors used in this study. At the least, this needs additional explanation so that this important element is understood.

Sorry for the confusion. Decarbonization in the power sector mainly impacts energy losses during electricity generation instead of T&D losses.

In the revised Introduction part, we clarified this as “the transition from fossil fuels to renewables in the power sector can substantially reduce energy losses during electricity generation” (Lines 60–62).

In the new section “Impacts of future changes on total source energy use”, we clarified that “The decarbonization process of the electric power sector mainly impacts primary energy consumption during electricity generation before delivering to buildings. On the one hand, switching from

coal-based power plants to renewable energy technologies reduces both greenhouse gas emissions and primary energy losses (Supplementary Table 10). On the other hand, the implementation of carbon capture and storage technologies can result in efficiency reductions and energy penalties^{26,27} (Lines 206–211). We further added the results of site-to-source conversion factors in Fig. 5 and Lines 214–228 to show the impact of decarbonization.

To establish baseline projections under different warming scenarios, we did not consider electrification of current on-site fossil fuel uses in this study. We clarified this in the revised Introduction (Lines 74–75) and Discussion (Lines 386–387).

In the “Conversion to source energy use” section in Methods, we mentioned that the changes in generation mix directly impact site-to-source building energy conversion factors, and explained in detail how the balancing area-level conversion factors were derived. This is also reflected in the newly added Supplementary Fig. 25 (which is a method overview).

The discussion section, where the policy implications are discussed, is a bit difficult to follow in places (paragraph 3 in particular). The policy implication discussion is primarily speculative or a bit too cursory to really understand the implications. It is often stated that the spatial heterogeneity in the response will require a “diversity of approaches” to meeting energy demand. But, it isn’t clear why that is truly needed and discussion of what those diverse approaches are is limited. The results indicate that population growth dominates the energy demand (the decarbonization may as well but as I have noted previously, I don’t understand that element of the paper), hence, it seems that the bulk impact of population and the spatial distribution of the increases will be important to the heterogeneous impact. However, the population dataset used is fairly coarse (1 km but downscaled from a 1/8 degree projection.... Or about 16 km at midlats I believe). Hence, my sense is that just keeping up with that demand will need simple capacity expansion in both the electricity generation and fossil fuel (NG) delivery. I am not sure what this means for diversity of approaches as I did not see a spatial breakdown of the population impact other than the latitudinally binned output (figure 6) which shows very little spatial heterogeneity.

Thanks for the comments and suggestions. We thoroughly revised the Discussion section, especially the first and third paragraphs, to enhance the policy implications. Please see Lines 306–396.

By “diversity of approaches”, we meant that spatial variations of climate change impacts must be considered when planning for urban energy infrastructure. Indeed, population growth seems to dominate future source energy use under most warming scenarios. This also partly explained why most existing grid projections are mainly based on population projections. However, simple capacity expansion in response to population growth while neglecting intracity variability induced by climate change (e.g., new Supplementary Figs. 19–20) might result in unmet energy demand in some urban areas. In the revised Discussion section, we argued that “considering population changes alone when making decisions on electric grid expansion in urban areas with similar population growth rates but distinct EUI changes (e.g., California) may lead to

underestimating or overestimating actual electricity demand. Therefore, the planning of electricity transmission, distribution, and storage infrastructure for each urban area must factor in these warming-induced variations. Likewise, the spatial variation in NE-EUI changes implies potentially varying requirements for pipelines and compressor stations to supply fossil fuels to buildings. This is especially the case for SSP3-7.0, under which the population-induced change is of similar magnitude to warming impacts on building fossil fuel use in many urban areas (Fig. 3 and Supplementary Fig. 23).” Please see Lines 311–320.

For the impact of decarbonization, please see our response to the previous comment.

For the spatial breakdown of the population impacts, the regional statistics are presented in Supplementary Figs. 21–22, and the distribution of population growth is presented in Supplementary Fig. 23. We are aware of the original resolution of this population dataset. However, it does have spatial variations among urban areas, as shown in Supplementary Figs. 23.

So, the wider impacts part of the paper – implications for energy supply capacity and resilience still seems a bit cursory or not as well thought out as the excellent work on building this methodology and getting these results. My impression looking at the results is that there will need to be significant capacity expansion of electricity generation everywhere with perhaps a bit more in the southern tier of cities. Since the grid is not generally “local” I don’t understand what the diversity of approaches to power supply means in this context. Certainly resilience in time as extreme events increase, that is clear. But, the system requirements for the combined impact of population/decarbonization/climate change seems to demand basic bulk increase in capacity..... hopefully renewable forms.

Thanks for the comments. Indeed, as mentioned above, the overall impact of these three drivers is increasing source energy use under most warming scenarios. However, this is not the case for SSP3-7.0, under which climate change plays an equally important role as population change and/or power sector change in many urban areas (e.g., Lines 290–294 and Figs. 6c and 7c).

Although the power grid is not generally local, each urban area has its own transmission and distribution power lines (or pipelines for natural gas). The spatial variability among urban areas implies that, in addition to simple population-induced capacity expansion, we also need to consider the variations induced by climate changes when planning for transmission, distribution, and storage infrastructure in individual urban areas. We clarified this in Lines 307–322.

My final recommendation is “accept with minor/major revision”. I extend into major because it may take work to build more analysis on to the population and decarbonization aspects of the results and build a better discussion section with clearer analysis/discussion on the wider implications to energy supply. However, that may not be difficult for the authors.... that is difficult to assess.

Thanks again for all the major comments. Please see our responses to specific comments in the tracked changed manuscript below.

Detailed comments in the track changed manuscript:

Line 5: “277 U.S. cities” - It is becoming common to refer to cities as “urban areas” and I believe it is the more apt phrase for the boundary definition you used.

We changed “cities” to “urban areas” in the revised manuscript and Supplementary Introduction. For consistency, we also changed our manuscript title to “Impacts of climate change, population growth, and power sector decarbonization on urban building energy use” (the journal allows up to 15 words for the title).

Line 9: “city-level” - It may be better to use “scale” or leave off altogether. In the manuscript there are times in which you may want to note that this is at the city scale. Other times, just using “city” alone is sufficient. “City-level” however, is often interpreted as a governance “level” and would be used if you were contrasting to state or federal levels of government. Here, however, you are not needing that contrast so either “scale” or “city” by itself seems more correct.

Thanks for the suggestion. We changed “city-level” to “city-scale” in the revision.

Line 12: “source energy use” - More on this later but I would suggest maybe some different wording on the source versus on-site.

Lines 49-53: Here I might recommend a different set of phrases to refer to these differing accounts of energy. This is just my opinion, but:

“Metered” energy consumption would refer to both electricity and NG/Petrol consumed in buildings. “site” isn’t the best because that often is interpreted as combustion of fuel on site, eliminating electricity from that phrase. “Metered” covers both and also clearly does not include transmission loss and so forth.

Then for the “primary” or “source” energy consumption, I would shift that to “total” energy consumption and define that here as you have - includes the losses etc.

I say this because I think I have seen “on-site” and “site” and sometimes just “building”. Again, just a preference, but at the least, be as clear and consistent as you can throughout.

Line 68: “For site energy use” - I won’t add metered to all of these. And perhaps you just define that unless you use the word “total”, it always implies the metered building scale.

Thanks for the great suggestions. We used the terms “site energy” and “source energy” mainly because: (1) both terms are more commonly used in building energy consumption and are also used by federal agencies such as the U.S. Environmental Protection Agency (e.g., U.S. EPA (2022)); and (2) they are also consistent with the variable “site-to-source conversion factor”, which is used in this study to convert electricity and fossil fuels consumed in buildings to primary energy use by considering energy losses during generation, transmission, and distribution.

We think “metered energy use” might lead to further confusion because of the following two reasons. (1) Although single-family homes in general have one electric meter and one gas meter per household, many apartment buildings often have one meter covering a wide range of units. This is even more complicated for commercial buildings, in which a single meter might cover multiple businesses, or a single commercial business might have multiple meters for their facility. Using “metered” energy use could give our readers the wrong impression that we are linking only one meter for each building. (2) As mentioned in Supplementary Method 5, (real) meter data are an important ground truth data source for the End-Use Load Profiles (EULP) database, while in this study, site energy use is modeled instead of measured. Using “metered energy use” to replace site energy use might create ambiguity.

We also did not use “total” for source energy use, because in section “Impacts of climate change on total energy use intensity” we used “total energy use intensity” to refer to the energy use intensity of both electricity and fossil fuels.

To avoid potential confusion, in the revised Introduction section, we pointed out that site energy is “electricity and fossil fuels consumed in buildings”, while source energy is site energy plus “energy losses during generation, transmission, and distribution” (Lines 58–59).

We also added “site energy use of buildings (hereafter simply “energy use”, unless otherwise specified)” in Line 77 and dropped “site” in the following sections to make the main text more concise.

In the revised Supplementary Information, we dropped “site” and clarified this in a new Supplementary Note 1: The use of “site” in the captions of supplementary figures.

Reference:

U.S. Environmental Protection Agency. (2022). EPA Recommended Metrics and Normalization Methods for Use in State and Local Building Performance Standards. 13 pages.
https://www.energystar.gov/sites/default/files/tools/BPS-Metrics_Recommendations_v7.pdf.

Line 48: “population dynamics and possible electric power sector changes” - I would imagine that beyond climate change, there are 4 additional factors: the two you mention here plus building technology changes and behavioral changes. Why are these not mentioned?

Thanks for pointing this out. The manuscript was initially transferred from a different journal which allows up to 3,000 words. To save space we only mentioned drivers/factors that are most

closely related to our analysis. In the revised Introduction section, we further introduced that changes in future urban building energy use are collectively determined by climate change, socio-economic developments (such as population growth and income), building technology advancements, possible changes in individual behavior, and electric power sector evolutions. Please see Lines 48–64.

Lines 69–70: “energy consumed in buildings divided by the total floor area of a city” - All buildings in a city?

Yes. This sentence has been moved to the revised Introduction section and becomes “final energy consumed in buildings divided by the total floor area of all buildings in an urban area”. Please see Lines 78–79.

Line 71: “city-level site electricity EUI” - I often use “E-EUI” and “NE-EUI” for electricity EUI and non-electricity EUI, respectively.

Thanks for this great suggestion. We used E-EUI and NE-EUI for electricity EUI and non-electricity EUI, respectively, in the revised manuscript. These two acronyms were defined in the last paragraph of the revised Introduction section: “Based on the types of final energy in buildings, it can be classified as electricity EUI (E-EUI) and non-electricity EUI (NE-EUI, including natural gas, fuel oil, and propane EUIs)” (Lines 80–82).

Line 75: “the 2010s level” - I know what you mean here, but this is awkward phrasing. Better to say “2010-2019 mean time period” or generate a defined shorthand first, then use (maybe “reference decade”).

Line 117: Same as previous comment.

We agree with the reviewer. In the revised “Impacts of climate change on annual electricity energy use intensity” section, we added “... during the period 2010–2019 (hereafter reference decade)” (Lines 90–91). We changed “the 2010s level” to “the reference-decade level” throughout the manuscript and Supplementary Information. We also changed “relative to the 2010s” in all figure captions (including supplementary) to “relative to the reference decade”.

Lines 75–76: “the change in annual total electricity EUI” - When looking at supplies figs 1 and 2 and then fig 1. It isn't clear to me if figure 1 is the sum of supporting fig 1 and 2 or whether it includes all electricity use....which could include water heating, lighting, devices, etc. In commercial buildings for example, there are a wide variety of electricity uses beyond space heating/cooling. Perhaps clarify this?

Sorry for the confusion. The results shown in Fig. 1 do include cooling, space heating, water heating, lighting, and all other miscellaneous electric end-use types. We clarified this in the revised Lines 93–95: “the change in annual total E-EUI, which includes space conditioning, water heating, and all other miscellaneous electric end-use types, varies substantially ...”.

Line 77: “The highest increases in annual electricity EUI” - Is this total or the metered?

This is the “metered” E-EUI (or site E-EUI). Following the convention defined in the revised Introduction, we dropped “site” and used “annual E-EUI”.

Line 87: “The latter factor leads to diverse trends” - Typically, “trend” makes me think of time. The figures are really spatial maps via the different SSPs. I have taken a stab at perhaps better wording here.

We agree with the reviewer. By “trend” we meant the directions of future changes (either increase/positive or decrease/negative). Following the reviewer’s suggestion, we changed this sentence to “The latter factor leads to diverse directions of annual E-EUI changes among urban areas ...” (Lines 105–106).

Lines 91–93: “These spatial disparities also suggest the diverse needs for future changes in electricity generation, transmission, and distribution to sustain urban energy systems.” Since this is explored primarily the discussion section, wait to make this point there.

Thanks for the suggestion. We moved this sentence to the discussion section.

Line 97: “building EUI” - I think you are referring to just electricity rather than for example, NG or petrol EUIs. The different hyphenated EUIs can solve that.

Thanks for pointing out this. Here we meant building EUI in general, including both E-EUI and NE-EUI. However, we only presented high-frequency E-EUI in the following paragraphs as it is more relevant to the resilience of power supply, especially during warm seasons and heat wave events.

Line 98: “Strong seasonal variations” - Throughout the manuscript, I would avoid these loaded terms. If it is a large number, use a numerical value to let the reader see the strength or weakness of a relationship or outcome. If you want to describe an outcome in this way use a word reflective of numerical outcomes: “large” or “small” rather than emotive terms like “strong”, “weak” “mild” etc.

Thanks for the suggestion. We changed these loaded terms to words reflective of numerical outcomes, such as “large” and “small”. We also changed “strong” to “(statistically) significant” when a significance test is possible. In addition, we changed “strong south–north gradients” to “consistent south–north gradients” in the revised abstract.

Lines 103–104: “both are much more intense” - I would avoid terms like “intense” unless truly referring to an intensity of some sort. It comes across as loaded language.

Following the reviewer’s suggestion, we changed this sentence to “both are much larger than the annual average change”. Please see Lines 118–119.

Line 111: “warm season” - Define warm seasons or just specifically refer to months.

The “warm season” was defined earlier in the same paragraph as “warm season (May–September)”. Please see Line 117.

Lines 122–123: “In particular, considerable increases in the number of peak hours (> 110% under SSP5-8.5) are projected in several coastal cities in California.” - Not sure about the relevance of this call-out? Perhaps clarify?

We meant to mention urban areas with the largest increase in peak hours. In the revision, we changed this sentence to “In particular, several coastal urban areas in California are projected to experience the largest increase in the peak demand frequency (e.g., > 110% under SSP5-8.5).” Please see Lines 136–137.

Line 129: “the site energy consumption ...” - Here I might say “non-electric metered building....”

Thanks for the suggestion. Here we meant the sum of electricity and non-electricity energy consumed in buildings, i.e., total (site) energy use in buildings. Following the convention defined in the revised Introduction, we changed “site energy consumption of buildings” to “building energy use” (Lines 143–144).

Line 133: “Fig. 3 and Supplementary Figs. 11 and 15” - Do you mean “to”?

Thanks for catching this. Yes, we meant “to”. We corrected this in the revision.

Lines 133–135: “Note that we do not explicitly model building technology change, so this is a product of warming-induced heat demand change, not efficiency improvement of fuel switching.” - Ahh. Here is the technology bit. This MUST be stated early on in the paper as many would consider this a fairly critical component of assessing projections of building energy demand under most circumstances. Indeed some general magnitude of this impact and some citations to literature would be appropriate. Furthermore, this is equally important to the electricity EUI so should not be only brought in this section.

We fully agree with the reviewer. In this revision, we highlighted in the very beginning (Introduction section) that “We hold building technologies, occupant behavior, and other socio-economic factors at their current levels, except for the AC penetration rate, and establish baseline projections for different future warming scenarios” (Lines 74–76).

We also added discussion regarding how building technologies and other factors might influence future building energy consumption in the revised Discussion section (Lines 371–385): “For example, a recent study⁴¹ found that aggressively improving building efficiency, such as deploying higher efficiency appliances and space-conditioning equipment, higher performance building envelope, and smart thermostats, can reduce electricity consumption in U.S. buildings by up to one quarter in 2050, assuming no changes in climate. The same study also estimated that full electrification of all buildings in the U.S. will lead to a one third increase in annual electricity consumption. Another modeling effort¹⁹ found that space heating electrification in Texas can reduce residential building energy use by 0–11%, although the magnitude depends on the efficiency assumed for heat pumps. These studies suggest the wide range of possible building energy use changes resulting from a variety of future building efficiency assumptions. Similarly, building energy saving also varies (by e.g., 4–30%) with different assumptions of occupant behavioral changes^{42,43}. Despite the important role that income plays in determining building energy use in low- and middle-income countries^{17,44}, several studies have suggested very low household income elasticity of residential building energy use in the U.S. when other building characteristics are controlled^{45,46}.”

We expect that these changes can provide a better context for the results of our study.

Please note that in all results sections we only mentioned this in the fossil fuel-related one. This is mainly because the impact of building technology changes on electricity use is more complicated than on fossil fuels (also suggested in the newly added discussion). The direction of changes will be largely determined by the end-use types considered. For example, replacing current space conditioning appliances with higher efficiency heat pumps will reduce space cooling energy use but increase space heating electricity use. This has been suggested in a recent study (White et al., 2021), which found that the change in annual residential electricity use ranges from 0% to 10.9% when replacing fossil-fuel furnaces with heat pumps in residential buildings in Texas. This large range is determined by the efficiency of heat pumps used in homes.

Reference:

White, P. R., Rhodes, J. D., Wilson, E. J., & Webber, M. E. (2021). Quantifying the impact of residential space heating electrification on the Texas electric grid. *Applied Energy*, 298, 117113.

Lines 138–141: “Owing to the largest share of fossil fuel consumption for space heating (e.g., Supplementary Fig. 16), cities in the Upper Midwest and Ohio Valley are projected to have the greatest reduction in annual site fossil fuel EUI in a warming future, with the maximum reduction ranging from 15.1% to 19.2% (Fig. 3).” - I think I understand what this sentence is intending to say, but it needs rephrasing. I think it is indicating that in the upper midwest and Ohio valley, space heating is the largest metered building fossil fuel energy consumption use in the residential and commercial buildings? Maybe you mean to say that “because space heating dominates the fossil use in these regions....”? Also, the share itself is a 2050s projected share and that seems important to also qualify here.

Line 139: “Supplementary Fig. 16” - This figure is only Natural Gas? Should this statement be qualified?

We revised this part based on the reviewer’s suggestion. Supplementary Fig. 16 (Supplementary Fig. 15 in the revision) is only for natural gas. We used natural gas as an example here because it is the dominant fossil fuel type for the vast majority of urban areas in the U.S. In the revision, we further quantified the share of space heating in (site) natural gas consumption during the reference decade and mid-century under all four scenarios to support our statement.

The revised text in Lines 157–162 is: “Space heating dominates urban building fossil fuel use in the Upper Midwest and Ohio Valley. For example, space heating on average accounts for 73.3–74.5% of mid-century total natural gas consumed in buildings across four SSPX-Y scenarios (Supplementary Fig. 15; cf. 77.4% during the reference decade). Therefore, urban areas in these regions are projected to have the greatest reduction in annual total NE-EUI in a warmer future, with the maximum reduction ranging from 15.1% to 19.2% (Fig. 3).”

Lines 141–144: “In comparison, cities in the southern part of the country have much milder changes in total fossil fuel EUI (e.g., reduction of <1.5% for most cities in Florida), despite the considerable relative changes in the fossil fuel EUI for space heating (Supplementary Fig. 15).” - This statement does not make sense to me. Or, at least, it just needs some clarification.

Sorry for the confusion. We revised this statement to make it clearer (Lines 162–165): “In comparison, urban areas in the southern part of the country exhibit much smaller changes in total NE-EUI (Fig. 3). For instance, despite the considerable changes in NE-EUI for space heating (Supplementary Fig. 14), the change in total NE-EUI is expected to be < 1.5% for most urban areas in Florida.”

Lines 146–147: “barring major end-use electrification policies” – “I would recommend saying, “barring other changes in non-space heating fossil fuel demand, such as end-use electrification policies”.”

Thanks for the suggestion. We found that following the reviewer’s suggestion this sentence would read: “Overall, non-space heating building fossil fuel energy demand accounts for an increasing share of city site fossil fuel consumption in the mid-century, barring other changes in non-space heating fossil fuel demand, such as end-use electrification policies”. The use of “non-space heating fossil fuel demand” sounds repetitive.

Therefore, we changed this sentence to: “Overall, non-space heating fossil fuels account for an increasing share of city-scale building fossil fuel consumption in the mid-century, barring major changes in the energy demand of non-space heating appliances, such as end-use electrification policies.” Please see Lines 165–168.

Line 147: This section makes no reference to support figs 12-14. Why? Surely the fuel oil figure deserves some comment as it is different from the others (presumably because fuel oil has a more diverse use beyond space heating)?

Thanks for pointing out this. We meant to cite these three supplementary figures in the first paragraph (we corrected this in the revision: “Supplementary Figs. 11–14”; Line 147). These three supplementary figures were also cited in the caption of Fig. 3.

We did not add detailed discussion in the original submission because (1) The results part was already dominated by climate change impacts; and (2) natural gas is the dominant fossil fuel type for most U.S. urban areas (> 96% urban areas), and the patterns of natural gas EUI changes is consistent with that for total NE-EUI.

The spatial patterns of fuel oil EUI and propane EUI are different from that for total NE-EUI because, compared with natural gas, the shares of fuel oil and propane for space heating are much more heterogeneous among U.S. urban areas.

In the revision, we added the following description in Lines 147–151: “Mid-century changes in city-scale natural gas EUI (Supplementary Fig. 11) are highly consistent with those for total NE-EUI, whereas changes in fuel oil and propane EUIs are geographically more heterogeneous (Supplementary Figs. 12–13). This difference is caused by the more diverse space-heating share in fuel oil and propane consumption among urban areas.”

Line 150: “site total EUI” - So does this mean elec + NG/petrol or does this mean including transmission etc losses? The language makes it a bit confusing.

As mentioned in the original Introduction section, “site” means final energy consumption in buildings, which does not include energy losses during generation, transmission, and distribution. In the revised Introduction section, we defined “site energy use of buildings

(hereafter simply “energy use”, unless otherwise specified)”. We also mentioned that EUI is only used for analyses of site energy use.

Here the “site total EUI” means electricity + fossil fuel consumption in buildings. To avoid confusion, we changed this phrasing to “total EUI of buildings, the sum of E-EUI and NE-EUI ...”. Please see Lines 171–172.

Line 153: “site fossil fuel EUI” - Again, I would recommend “metered building NE-EUI”

Thanks for the suggestion. We changed “site fossil fuel EUI” to “NE-EUI” in the revision.

Line 169: “local warming” - how is “local” defined?

By “local” we meant warming in each urban area (see “Linear regressions for the response of space-conditioning energy use intensity to warming” section in Methods. We clarified this in the revised Line 192: “warming in the urban area”.

Line 181: “electric power sector decarbonization” - It may be worth clarifying why decarbonization would impact energy since decarbonization is aimed at shifting away from carbon fuels but presumably delivering the same energy. I imagine this will certainly have an impact on T&D losses? But if there is more to it, worth a sentence or two here.

Thanks for the question. Electric power sector decarbonization mainly impacts energy losses during electricity generation instead of T&D losses. This is mainly because switching from fossil fuel-based power plants to renewables improves energy efficiency and reduces primary energy losses. In this study, this is reflected through changes in site-to-source conversion factor. A higher conversion factor means greater energy losses during electricity generation. We clarified this in Lines 206–213: “The decarbonization process of the electric power sector mainly impacts primary energy consumption during electricity generation before delivering to buildings. On the one hand, switching from coal-based power plants to renewable energy technologies reduces both greenhouse gas emissions and primary energy losses (Supplementary Table 10). ... Here we consider two possible futures of the U.S. electric power sector, and evaluate building source energy consumption through site-to-source conversion factors (Methods; Fig. 5).”

As for T&D losses, we adopted a constant grid gross loss of 5% based on U.S. EPA’s eGRID data (the 2010s) to account for T&D losses. We mentioned this in Supplementary Method 6. Although the power sector decarbonization process could also potentially influence T&D losses, we noticed that in the projections from the Cambium dataset, a constant distribution loss rate of 3.6% is used for the entire study period, while the national transmission loss rate slightly increases from 0.9% to 1.0% during the period of 2022–2050. As a result, the national T&D loss rate is projected to change from 4.5% in 2022 to 4.6% in 2050. Although this loss rate is slightly lower than eGRID, it does support the assumed constant T&D losses in this study. Considering

that the grid gross loss from eGRID is derived from historical data, we decided to use 5% instead of 4.5% in our analysis.

Lines 191–198: “As revealed by the logarithmic mean Divisia index method ... low share of coal (Supplementary Table 2).” - The reality of Figure 6 is obviously quite important. Because the paper’s topline emphasis seems to be more about the impact of climate change..... the fact that it is quite small compared to these other two factors (and of course there is the building tech change, policy changes, behavioral changes), may appear to be a somewhat unbalanced presentation of the results? I would recommend a bit more balance up front, so the reader can quickly appreciate the relative magnitude of the three drivers analyzed here.

Good point. We completely agree with the reviewer. In this revision, the following two major changes were made to balance the results of climate change impacts and those of population growth and power sector decarbonization:

(1) We added a new Fig. 5, site-to-source conversion factors for electricity, to illustrate the impact of power sector decarbonization on source energy use. This figure will help the readers understand the conversion from site energy use to source energy use of buildings. We also moved the original Supplementary Fig. 19 to the main text to supplement the original Fig. 5. The new Figs. 6 and 7 provide a more complete picture of source energy consumption changes under all four warming scenarios and two power sector decarbonization scenarios. With these new figures, the revised manuscript now has four figures on site energy use (mainly climate change impacts) and four on source energy use (all three factors).

Fig. 5 | National and regional site-to-source conversion factors for electricity in the 2010s and 2050s with electric power sector decarbonization under the business-as-usual and zero-carbon scenarios. Error bars represent the variability (± 1 standard deviation) among urban areas within each region. Conversion factors in the 2050s for each city are averaged across four SSPX-Y scenarios.

(2) We added more description on the impacts of power sector decarbonization and population growth. More specifically, we split the original “Source energy use change and attributions” section into two new sections: “Impacts of future changes on total source energy use” and “Attributions of source energy use change”. In the “Impacts of future changes on total source energy use” section, we described in detail how population growth and power sector decarbonization impact source energy consumption. In the “Attributions of source energy use change” section, we described the patterns under all four SSPX-Y scenarios and two power sector scenarios, and further explained the observed south–north gradient. Note that we also added two new Supplementary Figs. 19 and 20 to show the contribution of climate change to source energy use changes.

Following the reviewer’s suggestion, in the very beginning of the revised section “Attributions of source energy use change”, we mentioned that “as revealed by the decomposition, under all SSPX-Y scenarios except SSP3-7.0, population growth is the primary driver of future source energy use changes for most urban areas, followed by power sector change and climate change”. See Lines 250–253.

Lines 199–200: “direct air capture, carbon capture and sequestration, and the complete phase-out of coal use” - These will impact CO₂ emissions, but why would they impact the energy?

The use of carbon capture and storage directly affects heat rates of source energy types and therefore changes energy losses during electricity generation. This is suggested in Supplementary Table 10. We acknowledge that phase-out of coal use decreases primary energy losses whereas carbon capture and storage technologies increase primary energy losses. This was clarified in Lines 210–211: “the implementation of carbon capture and storage technologies can result in efficiency reductions and energy penalties^{26,27}.”

To avoid confusion, we changed this sentence to “More aggressive power sector decarbonization, with the complete phase-out of coal use, could further compensate ...” Please see Lines 270–271.

Line 202: “net-zero power sector” - How? Is this a feedback: reduce emissions which then softens the climate change impacts which then leads to less energy demand for cooling?

This is mainly through the reduced energy losses during electricity generation. We clarified this in the revised Introduction, Section “Impacts of future changes on total source energy use”, and Section “Attributions of source energy use change”. Please see our responses to the comments above.

In the Methods section, we also mentioned that “The combination between power sector scenarios and SSPX-Y scenarios enables us to probe the sensitivity of building source energy consumption in U.S. urban areas to global climate change, in which the regional power sector

scenarios are not informed by the global SSPX-Y scenarios”. Therefore, this is not a feedback effect.

Lines 214–216: “Climate change contribution under SSP3-7.0 can even be the same order of magnitude as that of population change, which completely offsets the source energy increase induced by population growth in some regions.” - ? This seems contradictory to the previous text?

Sorry for the confusion. The previous text in the original submission, “The contribution of climate change to projected changes in building source energy use is smaller than that of population growth or power sector decarbonization”, is based on *average changes*. This is mainly because the projected increases and decreases in total building energy use are canceled out when averaged across all urban areas. However, the maximum climate change-induced increase and decrease in city-scale source energy consumption are much larger than the average change. Under the SSP3-7.0 scenario, when the population growth is the smallest, climate change impacts can be greater than population impacts for several urban areas in the Northeast and Upper Midwest. In the revision, we clarified this in Lines 283–294:

“When averaged across all urban areas, climate change is projected to reduce city-scale building source energy use by 2.5%, 2.1%, 2.0%, and 2.4% under SSP1-2.6, SSP2-4.5, SSP3-7.0, and SSP5-8.5, respectively (Supplementary Figs. 19–20). The relatively small mean contribution of climate change, compared with that of population growth or power sector decarbonization, is partly because global warming causes total building energy use increases in some urban areas while decreases in others (Fig. 8). In fact, the maximum climate change-induced increase and decrease in city-scale source energy consumption are projected to be ~3–8% and ~8–11%, respectively, under different warming scenarios. Especially, the contribution of climate change under SSP3-7.0 is the same order of magnitude as that of population change for some urban areas in the Northeast, Upper Midwest, and Ohio Valley, where the source energy use increase caused by population growth can be largely or even completely offset by the warming-induced reduction.”

Lines 219–222: “The high spatial granularity of our approach reveals highly heterogeneous response of city-level building EUI to the mid-century climate change, indicating potential diverse needs for the deployment of future urban energy infrastructure.” - This is a seemingly central policy implication of the work as presented. However, it seems to be more of an assertion for which the linkage to the results are not clear. There is a heterogeneous set of changes in demand, for sure. But, existing building energy demand is already diverse spatially so why would a different future heterogeneity require system change or planning? I get the argument about resilience vis a vis the increase in extremities, but the argument for greater energy infrastructure needs more evidence or literature support. Otherwise, it seems like a hand-waving argument. Numerically, planning for the projected changes in the coming decades seems more dependent

on population and building technology.... The change induced by climate change seems relatively small so not sure why it is critical to energy delivery?

Thanks for the suggestions and sorry for the confusion. The major implication in this paragraph is that we need to consider the heterogeneous impacts of climate change on top of population and power sector changes. Although on average the impact of climate change seems small, the warming-induced building E-EUI change can range from -4% to 7% in individual cities. This spatial heterogeneity is rarely reported in the literature. Currently, many projections of grid expansion only consider the impacts of socioeconomic developments such as population growth (e.g., U.S. EIA projections). Therefore, we suggest here that the warming-induced spatial heterogeneity should also be considered in future planning of urban energy delivery. In the revision, we clarified this and mentioned key results to support our statement. Please see the revised Lines 307–322:

“The high spatial resolution of our approach reveals heterogeneous city-scale building EUI changes in response to future warming (Supplementary Fig. 16). Depending on building stock characteristics, urban areas in the same state or even the same county will have varying future changes in building EUI under different warming scenarios (Figs. 1 and 3). Consequently, the need for future urban energy infrastructure deployment will differ among urban areas. For instance, considering population changes alone when making decisions on electric grid expansion in urban areas with similar population growth rates but distinct EUI changes (e.g., California) may lead to underestimating or overestimating actual electricity demand. Therefore, the planning of electricity transmission, distribution, and storage infrastructure for each urban area must factor in these warming-induced variations. Likewise, the spatial variation in NE-EUI changes implies potentially varying requirements for pipelines and compressor stations to supply fossil fuels to buildings. This is especially the case for SSP3-7.0, under which the population-induced change is of similar magnitude to warming impacts on building fossil fuel use in many urban areas (Fig. 3 and Supplementary Fig. 23). These intercity variations emphasize the location-dependent challenges and opportunities each urban area will face for energy saving and energy efficiency improvement.”

Lines 223–224: “processing facilities and pipelines for fossil fuels” - This is the first mention of this?

To enhance the connection to our results, we changed this to “the spatial variation in NE-EUI changes implies potentially varying requirements for pipelines and compressor stations to supply fossil fuels to buildings.” Please see Lines 316–318.

Line 239: “global mean temperature changes” - This seems like a straw man argument... there are studies that focused on regional/national scales that used gridded model output to drive exploration of energy impacts.

We fully agree with the reviewer. The “global mean temperature changes” mentioned here actually refers to previous studies that derived the linear response of energy use to warming (Hsiang et al., 2017; Li et al., 2019). Similar to IPCC climate assessment reports (e.g., IPCC (2022)), these studies mainly use global surface temperature change when describing the linear response. In comparison, our study provided city-specific linear response and can benefit urban planning and policy-making. We revised this sentence to “Previous studies have used global mean temperature changes to quantify the linear or near-linear responses of energy use to future warming^{3,25}. In comparison, here we derive city-scale responses of space-conditioning EUI to local air temperature changes ...” to avoid confusion. Please see the revised Lines 336–338.

References:

Hsiang, S., Kopp, R., Jina, A., Rising, J., Delgado, M., Mohan, S., Rasmussen, D. J., Muir-Wood, R., Wilson, P., Oppenheimer, M., Larsen, K. & Houser, T. (2017). Estimating economic damage from climate change in the United States. *Science*, 356(6345), 1362–1369.

IPCC (2022). Climate Change 2022: Impacts, Adaptation and Vulnerability. Contribution of Working Group II to the Sixth Assessment Report of the Intergovernmental Panel on Climate Change. Cambridge University Press. Cambridge University Press, Cambridge, UK and New York, NY, USA, 3056 pp.

Li, Y., Pizer, W. A., & Wu, L. (2019). Climate change and residential electricity consumption in the Yangtze River Delta, China. *Proceedings of the National Academy of Sciences of the United States of America*, 116(2), 472–477.

Lines 241: “The impact of urban heat mitigation and adaptation strategies ... better evaluate their cost-effectiveness.” This is hard to follow. There seem to be a variety of new policy concepts introduced here and unclear what “readily convert” means. This is the first introduction of cost-effectiveness. While there may be useful and important implications here, they are brought in without any background or general support but seem mostly speculative.

Sorry for the confusion. We thoroughly revised this paragraph to make it clearer. Please see Lines 336–355:

“Previous studies have used global mean temperature changes to quantify the linear or near-linear responses of energy use to future warming^{3,25}. In comparison, here we derive city-scale responses of space-conditioning EUI to local air temperature changes. These linear responses are informative for urban planning and policy-making processes, particularly in the development of strategies for urban heat mitigation and adaptation. To improve urban sustainability, global cities are actively developing and implementing strategies such as green infrastructure and cool materials to reduce heat stress and building energy use in warm seasons^{32,33}. The city-scale cooling effect of heat mitigation strategies can be quantified reasonably well by mesoscale meteorological models. However, the effectiveness of these strategies in reducing building energy use is conventionally evaluated on a case-by-case basis at the building- or community-scale^{34–36}. Considering the potential tradeoff between cooling energy savings and space heating

penalties as well as distinct building stock characteristics across urban areas, the resulting city-scale energy use change is often uncertain^{37–39}. This is partly reflected in some urban areas along coastal California, where cooling and space heating are both highly sensitive to temperature change (Fig. 4). The city-scale responses quantified herein (Fig. 4 and Supplementary Fig. 17) provide a solution to this issue for urban planners and decision-makers. For instance, when assessing the impact of specific heat mitigation strategies in an urban area, using the quantified linear response, city-scale changes in space-conditioning building energy use can be conveniently estimated from the predicted temperature change (e.g., by meteorological models).”

Lines 249–251: “Under all warming scenarios except SSP3-7.0, the warming-induced change in source energy consumption for urban buildings is one order of magnitude smaller than that governed by socio-economic forces.” - A fairly important admission. What socio-economic forces are you referring to here? Is this referring to the population and power sector changes shown in Figure 6?

The socio-economic forces here refer to population growth, an important indicator of socio-economic development. We changed “socio-economic forces” to “population growth” to avoid confusion. Please see Line 358.

Line 251: “increasing trend” - What trend are you referring to? The trend associated with the dominance of socio-economic forces from the previous sentence? Also an increasing trend implies and acceleration?

The “increasing trend” here means the increase in source energy use. We changed “increasing trend” to “projected rises in source energy use” in Line 358.

Lines 252–254: “CO2 emissions from the remaining fossil fuel-based power plants under business-as-usual scenario can be eliminated through decommissioning and negative emission technologies as under zero-carbon scenario.” - The referral to decarbonization seems odd within the context of the paper. Changing climate lowers source energy use, by and large (figure 6) to small degree - the increase in source energy requirements are driven mainly by population.... Decarbonization lowers future building energy demand (figure 6) though I still do not understand how that works. But this statement moves to CO2 reduction.... Which seems minimally related to the building energy consumption. On other words, the introduction of CO2 reductions seems like a separate topic from the aims of the paper here

Thanks for the suggestion. We discussed CO₂ emission reductions here because we intended to highlight the importance of considering electrifying fossil fuel-based appliances in future studies, which is closely related to CO₂ emissions. We revised this paragraph to highlight the connection between energy consumption and CO₂ emissions (Lines 358–369):

“The projected rises in source energy use under most warming scenarios (Figs. 6–8) suggest potential changes in associated CO₂ emissions, signaling the great challenges toward achieving decarbonization. While fossil fuel-based power plants under the business-as-usual scenario still emit CO₂ when supplying electricity to buildings, their emissions can be offset through decommissioning and negative emission technologies (e.g., carbon capture and storage and direct air capture) under the zero-carbon scenario. However, with the growing population, direct fossil fuel combustion in buildings is projected to increase in almost all urban areas under SSP1-2.6, SSP2-4.5, and SSP5-8.5 and most urban areas under SSP3-7.0 (Supplementary Fig. 24). Correspondingly, their associated CO₂ emissions remain a key barrier to decarbonization. Therefore, rapid electrification of future urban buildings is of equal importance to power sector decarbonization⁴⁰, particularly under higher warming/emission scenarios. We note this as an important area for future study.”

For the impact of decarbonization on source energy use, it is mainly through the reduction of energy losses during electricity generation. Please see our responses to your comments above.

Lines 260–263: “It is noteworthy that changes in ... well understood from survey data.” This is ok as a caveat. But, I think it would be more comprehensive for the reader to be shown all the potential categories of drivers of building energy demand and then note what is considered in this study and what is not. Then, here where the caveats are being made, I would find some sort of literature support for roughly how influential these other factor are. Otherwise the study leaves the reader with the wrong impression and limited context to place the changes analyzed here within the bigger context of building energy demand in the future.

Thanks for the suggestions. In the revised Introduction, we added new content to introduce all potential drivers of building energy demand following the reviewer’s suggestion, and highlighted that we only considered three drivers in this study (Lines 37–64 and 74–76).

In this paragraph, we further added literature and discussion to estimate the potential impacts of other factors. Please see Lines 370–386:

“Changes in building construction and operation, building codes, appliance efficiency, and other economic factors such as income and price can also influence building EUI^{7,9,24}. For example, a recent study⁴¹ found that aggressively improving building efficiency, such as deploying higher efficiency appliances and space-conditioning equipment, higher performance building envelope, and smart thermostats, can reduce electricity consumption in U.S. buildings by up to one quarter in 2050, assuming no changes in climate. The same study also estimated that full electrification of all buildings in the U.S. will lead to a one third increase in annual electricity consumption. Another modeling effort¹⁹ found that space heating electrification in Texas can reduce residential building energy use by 0–11%, although the magnitude depends on the efficiency assumed for heat pumps. These studies suggest the wide range of possible building energy use changes resulting from a variety of future building efficiency assumptions. Similarly, building energy saving also varies (by e.g., 4–30%) with different assumptions of occupant behavioral changes^{42,43}. Despite the important role that income plays in determining building energy use in

low- and middle-income countries^{17,44}, several studies have suggested very low household income elasticity of residential building energy use in the U.S. when other building characteristics are controlled^{45,46}.”

Line 291: “to release” - ??

Sorry for the confusion. We changed this sentence to “PUMAs are the lowest level of geography used by the U.S. Census Bureau for the tabulation and dissemination of the American Community Survey (ACS) Public Use Microdata Sample data.” Please see Lines 419–421.

Line 301: “we included 277 cities” - What dictated this number? Not clear to me.

We revised this part to improve the clarity: “To reconcile the mismatch between urban areas and PUMAs, we identified one or more PUMAs to represent each urban area. More specifically, a PUMA is selected if at least 10% of its spatial extent is co-located with an urban area (hereafter representative PUMAs). On the one hand, this threshold ensures that we can cover a sufficiently large number of CONUS urban areas. On the other hand, the selection also covers part of the urban periphery which may undergo future urban development and potential population growth. In cases where a PUMA extends over multiple urban areas, we assigned it to the urban area that encompasses the largest proportion of the PUMA’s spatial extent to avoid double counting. Following these procedures, we excluded urban areas without any representative PUMAs and retained 277 out of 481 CONUS urban areas in this study. These 277 urban areas cover 1,540 PUMAs across the country (Supplementary Figs. 26–27).” Please see Lines 422–433.

Lines 387–388: “water mains temperature” - Maybe “ambient water temperature”?

Thanks for the suggestion. Water mains temperature here is the temperature of water mains that deliver water to a building via underground pipes. It is a function of outdoor climate conditions with temporal variations. Therefore, water mains temperature is different from ambient water temperature.

Line 411: “building stock energy consumption” - Some methods flow-type diagrams would be very helpful for this manuscript. They can be in supplementary information, but the methods are dense with so many related states a diagram would be critical.

Thanks for the suggestion. We added a new Supplementary Fig. 25 to provide an overview of the bottom-up modeling approach developed in this study.

[redacted]

Supplementary Fig. 25. Overview of the hybrid bottom-up urban building energy modeling approach developed in this study and main data sources.

Line 451: “with coefficients absorbed” - ??

We meant that, for simplicity, the regression coefficients for these fixed effects variables were omitted. Given that it is fairly common to omit regression coefficients for these three sets of variables, we removed “, with coefficients absorbed” to avoid confusion.

Fig. 1: It is not clear to me if this is the sum of supplementary figures 1 and 2 or if this reflects EUI for all electricity consumption. Important as there are many other uses of electricity in both residential and commercial buildings.... Particularly commercial buildings. These would be quite different metrics. Please clarify.

This comes up everywhere in the manuscript when the “site EUI” is used.

Sorry for the confusion. Fig. 1 shows electricity EUI (or E-EUI in the revision), which is the sum of all electric end-use types (Supplementary Fig. 1 + Supplementary Fig. 2 + all other miscellaneous types). In the revised section “Impacts of climate change on annual electricity energy use intensity”, we clarified this as “... the change in annual total E-EUI, which includes space conditioning, water heating, and all other miscellaneous electric end-use types, varies substantially among urban areas across different warming scenarios (Fig. 1).” Please see Lines 93–95.

Please note that we omitted “site” in EUIs in the revision.

Fig. 2: Include number of cities in each region as was done in supplementary figure 3

Thanks for the suggestion. We revised Figure 2 to include the number of cities in each climate region. We also changed the y-axis title from “Change of hourly on-site electricity use intensity (%)” to “Change in hourly electricity energy use intensity (%)” for consistency. Please see the revised Figure 2 below.

Fig. 2 | National and regional changes in hourly electricity energy use intensity in the 2050s relative to the reference decade when averaged over the annual scale, the warm season, and the top 5% hottest days. In each subplot, the left, middle, and right panels show changes in the annual average, the average of the warm season (May–September), and the average of the top 5% hottest days over the simulated decade, respectively, based on the ensemble mean of the simulations driven by 10 CMIP6 models under each SSPX-Y scenario. Error bars represent the variability (± 1 standard deviation) among urban areas within each region. N in each subplot denotes the number of urban areas. The division of nine climate regions is shown in Supplementary Fig. 3.

Fig. 5: I would like to see this figure without pop change or the decarbonization... just the climate change induced change in total source energy. The next figure helps, but the isolate of CC induced energy change at the scale of the cities is needed.

Isolating climate change-induced source energy use change at the city scale is possible with the help of LMDI decomposition, just like what we did in Fig. 8 (originally Fig. 6). In the revision, we added two new figures in the supplementary, Supplementary Figs. 19 and 20, to show the relative change in source energy use attributable to climate change under business-as-usual and zero-carbon scenarios. Please see below for the results under business-as-usual scenario as an example. As LMDI is a perfect decomposition, the results shown in these two new supplementary figures are nearly identical. This also suggests the robustness of the decomposition method.

Supplementary Fig. 19. Climate change-induced annual source energy consumption change in the 2050s relative to the reference decade under four illustrative SSPX-Y scenarios and the business-as-usual power sector decarbonization scenario. a, SSP1-2.6 scenario. b, SSP2-4.5 scenario. c, SSP3-7.0 scenario. d, SSP5-8.5 scenario. Each point represents the relative change (%) based on the ensemble mean of the simulations driven by 10 CMIP6 models under each SSPX-Y scenario.

Supplementary Fig. 20. Climate change-induced annual source energy consumption change in the 2050s relative to the reference decade under four illustrative SSPX-Y scenarios and the zero-carbon power sector decarbonization scenario. a, SSP1-2.6 scenario. b, SSP2-4.5 scenario. c, SSP3-7.0 scenario. d, SSP5-8.5 scenario. Each point represents the relative change (%) based on the ensemble mean of the simulations driven by 10 CMIP6 models under each SSPX-Y scenario.

Fig. 6: As earlier comments note, this figure makes it clear that the climate change impact is small relative to the apparently far more important changes in population and decarbonization (though I still don't understand how this has such a large impact on demand other than T&D loss).

Thanks again for the comment. As mentioned above, the *average* impact of climate change is small, but the impact in individual urban areas can be the same order of magnitude as population changes. We made several changes in this revision to make this clear. In addition, decarbonization changes source energy use through primary energy losses during electricity generation. Please see our detailed responses to the previous comments.

All other edits in the tracked changes manuscript file.

Thanks again for all the other edits in the tracked changes document, which greatly improved our manuscript. We incorporated these changes and thoroughly revised the manuscript to improve coherency and clarity. All changes have been highlighted in the tracked changes version of the manuscript.

Responses to Reviewer #2:

We appreciate the insightful and constructive comments from Reviewer #2. Please see our point-by-point responses below. Note that the line numbers are based on the word document, not the PDF file converted by the submission system.

1. The terminology for the number of peak hours seems confusing. The peak hours normally refer to the time when HVAC demand reaches maximum. The increase of 170% is substantial and could be misleading. Recommend the authors present the absolute number of hours as well.

Thank you for the suggestion. Peak hours in this study are defined as “hours with electricity EUI higher than the current (the reference decade) 95th percentile”. We analyzed the number of peak hours in this study as an important indicator of peak energy use *frequency* (cf. *intensity* in Fig. 2). Similar definitions have been used in previous studies. For example, Auffhammer et al. (2017) reported up to 406% increase in peak load frequency (number of peak days) under RCP8.5 for the Electricity Reliability Council of Texas region by the end of the 21st century.

To avoid confusion, based on both reviewers’ suggestions, we changed “up to 170% increases in the number of peak hours for electricity EUI” to “up to a 170% increase in the frequency of peak building electricity EUI” in the abstract. We clarified in the main text that “To examine the change in peak demand frequency, we define peak hours as hours with E-EUI [electricity EUI] higher than the current (the reference decade) 95th percentile” and “the relative change in the peak demand frequency (number of peak hours) under different warming scenarios” (Lines 129–132). We also clarified in the revised Supplementary Fig. 10 that “the number of peak hours is used to indicate peak demand frequency”.

Another important reason we used this definition is that readers can quickly convert the relative change to the absolute change in the number of peak hours. Based on the definition, the number of hours with electricity EUI higher than the current (the reference decade) 95th percentile is $5\% \times 8760 \text{ hours/year} \times 10 \text{ years} = 4380 \text{ hours}$ during the decade 2010–2019. Therefore, if an urban area is projected to have a 170% increase in the number of peak hours (or peak demand frequency) under SSP5-8.5, the number of peak hours in this urban area will become $4380 \times (1 + 170\%) = 11826 \text{ hours}$ during the decade 2050–2059. To help our readers better understand the relationship between these two scales, we added the absolute number of hours in the revised Supplementary Fig. 10, and mentioned in the caption that “based on this definition, the number of peak hours for each urban area during the reference decade is 4380”.

Reference:

Auffhammer, M., Baylis, P., & Hausman, C. H. (2017). Climate change is projected to have severe impacts on the frequency and intensity of peak electricity demand across the United States. *Proceedings of the National Academy of Sciences of the United States of America*, 114(8), 1886–1891.

Please see the revised Supplementary Fig. 10 below:

Supplementary Fig. 10. Change in the number of peak hours for electricity energy use intensity in the 2050s relative to the reference decade under four illustrative SSPX-Y scenarios. A, SSP1-2.6 scenario. B, SSP2-4.5 scenario. C, SSP3-7.0 scenario. D, SSP5-8.5 scenario. Each point represents the relative change (%) based on the ensemble mean of the simulations driven by 10 CMIP6 models under each SSPX-Y scenario. Here the number of peak hours is used to indicate peak demand frequency. Peak hours are defined as hours with electricity energy use intensity higher than the current (the reference decade) 95th percentile of electricity energy use intensity. Based on this definition, the number of peak hours in each urban area during the reference decade is 4380.

2. For each 1°C warming, city-level space-conditioning EUI on average rises and drops by 14% and ~10% for cooling and space-heating, respectively. The increase of 14% in cooling was only mentioned in the abstract. The analysis conducted in this study seems insufficient to draw this conclusion. Did the research team simulate and test the impacts of every 1°C interval within a temperature range on cooling and heating energy use?

Thanks for the question. We would like to point out that “the increase of 14% in cooling” was also mentioned in Section “City-scale response of energy use intensity for space-conditioning to warming”: “On average, city-scale building E-EUI for cooling rises by 13.8% for each 1°C ...”. We rounded 13.8% to 14% in the abstract for consistency. This 13.8% increase (or 14% in the abstract) is the average of all urban areas across the country as shown in Fig. 4a.

These city-scale space-conditioning EUI responses are not based on the evaluations for every 1°C interval. Instead, they are based on the linear regression models between annual building cooling/space heating demand and city-scale annual temperature change. This method has been used in previous studies (e.g., Hsiang et al., 2017; Li et al., 2019) although with global mean surface temperature change. In our study, the linear regression model for each urban area is constructed based on 40 mean annual temperature changes (10 CMIP6 climate models \times 4 SSPX-Y scenarios) and 40 corresponding space-conditioning EUI changes. We clarified this in the Methods part, section “Linear regressions for the response of space-conditioning energy use intensity to warming” (Lines 685–692): “we observed significant linear dependence of both building cooling demand and space heating demand on warming in our study. For each urban area, the projected changes in annual cooling demand, space heating demand, and air temperature from a wide spectrum of 40 potential trajectories (10 CMIP6 climate models \times 4 SSPX-Y scenarios) provide a sufficiently large sample size to build linear regression models (Supplementary Fig. 31). Of particular interest is the slope of the linear regression model, which indicates the local response of the cooling or space-heating EUI to a city-scale temperature change of 1°C (for relative change the unit is %/°C).” Two city-scale examples were also shown in Supplementary Fig. 34 (copied below).

Supplementary Fig. 34. Examples of linear relationships between city-scale warming magnitude and change in annual electricity energy use intensity for cooling in the 2050s

relative to the reference decade based on the projections of 10 CMIP6 models under four illustrative SSPX-Y scenarios. a, The response of electricity energy use intensity for cooling to local warming in Houston, Texas with fixed air-conditioning saturation rate as in the 2010s. **b,** The response of electricity energy use intensity for cooling to local warming in Houston, Texas with changing air-conditioning saturation rate. **c,** The response of electricity energy use intensity for cooling to local warming in Seattle, Washington with fixed air-conditioning saturation rate as in the 2010s. **d,** The response of electricity energy use intensity for cooling to local warming in Seattle, Washington with changing air-conditioning saturation rate. Black lines represent linear fits to 40 data samples (10 climate models \times 4 SSPX-Y scenarios). The p -value is calculated based on the F -test. Note that the change in air-conditioning saturation rate varies in different climate models. EUI in this figure represents energy use intensity.

We also evaluated the robustness of these linear regression models. As mentioned in the Methods part (Lines 693–704), “We fitted linear regression models between the city-scale air temperature change and mid-century changes in E-EUI for cooling, E-EUI for space heating, natural gas EUI for space heating, fuel oil EUI for space heating, and propane EUI for space heating with the ordinary least squares method. The linear regression models for these five space-conditioning EUIs are statistically significant for all urban areas (p -value $< 10^{-8}$). The minimum R^2 values for the responses of E-EUI for cooling, E-EUI for space heating, natural gas EUI for space heating, fuel oil EUI for space heating, and propane EUI for space heating are 0.757, 0.666, 0.673, 0.716, and 0.596, respectively. In addition, the warming-response of E-EUI for space cooling with AC saturation rate fixed at the reference-decade level exhibits a spatial pattern similar to that for the case with changing AC saturation rate (Fig. 4a and Supplementary Fig. 17a), although with a relatively smaller magnitude of response. This suggests the robustness of responses quantified using linear regression models (e.g., Supplementary Fig. 34).”

References:

- Hsiang, S., Kopp, R., Jina, A., Rising, J., Delgado, M., Mohan, S., Rasmussen, D. J., Muir-Wood, R., Wilson, P., Oppenheimer, M., Larsen, K., & Houser, T. (2017). Estimating economic damage from climate change in the United States. *Science*, 356(6345), 1362–1369.
- Li, Y., Pizer, W. A., & Wu, L. (2019). Climate change and residential electricity consumption in the Yangtze River Delta, China. *Proceedings of the National Academy of Sciences of the United States of America*, 116(2), 472–477.
-

3. What are the resources of the measurement data for statistical calibration?

Thanks for the question. The statistical calibration uses data from the End-Use Load Profiles (EULP) database, and mainly limited by the length requirement of the journal, we only briefly mentioned major data sources in Methods (Lines 543–545): “... the U.S. Energy Information Administration’s Residential Energy Consumption Survey (RECS) and Commercial Buildings Energy Consumption Survey (CBECS), and the U.S. Census Bureau’s American Housing Survey

(AHS) and ACS”. Nevertheless, the measurement data sources (metered and/or surveyed) are summarized in Supplementary Method 5, which is copied here:

“... major data sources used to develop representative residential building models in ResStock⁸ include the U.S. Census Bureau 2012–2016 American Community Survey (ACS) 5-year data³⁰, U.S. Census Bureau 2013–2017 ACS 5-year Public Use Microdata Sample³¹, U.S. Energy Information Administration (EIA) 2009 Residential Energy Consumption Survey (RECS) data³², 2015 RECS data³³, U.S. Census Bureau 2017 American Housing Survey (AHS) data³⁴, Northwest Energy Efficiency Alliance (NEEA) Residential Building Stock Assessment (RBSA) data (2011–2012)³⁵, NEEA RBSA II data (2016–2017)³⁶, 2009 International Energy Conservation Code (IECC)³⁷, ANSI/RESNET/ICC 301-2019 Standard³⁸, Homeland Infrastructure Foundation-Level Data (HIFLD) national parcel database (not available to the public), and the 2014 Building America House Simulation Protocols³⁹”

“Major data sources used to construct a network of conditional probability distributions of building characteristics for ComStock⁸ include U.S. Census Bureau 2013–2017 ACS 5-year Public Use Microdata Sample³¹, U.S. EIA 2012 Commercial Buildings Energy Consumption Survey (CBECS) data⁴², Homeland Security Infrastructure Program (HSIP) Gold 2012 database, 2017 CoStar Real Estate Data⁴³, 2015 U.S. Lighting Market Characterization⁴⁴, ASHRAE Service Life and Maintenance Cost Database⁴⁵, NEEA Commercial Building Stock Assessment (CBSA) 4 (2019)⁴⁶, ANSI/ASHRAE Standard 62.1 Ventilation for Acceptable Indoor Air Quality, ANSI/ASHRAE/IES Standard 90.1 Energy Standard for Buildings Except Low-Rise Residential Buildings, Building Codes Assistance Project⁴⁷, the Database for Energy Efficient Resources (DEER)⁴⁸, commercial prototype building models^{27,28}, and different survey and submeter data.”

“A large amount of empirical ground truth data (both public and private) have been acquired to facilitate region-by-region calibration and multi-dimensional, multi-variable validation of the EULP database. For residential buildings, 18 data sources were used for calibration and validation, including national and state-level survey data on annual and monthly scales, regional and city-scale end-use metering data on the hourly scale, regional load research data on the hourly scale, and eight regional or city-scale hourly advanced metering infrastructure (AMI) datasets. The calibration and validation of commercial buildings also used 18 data sources, including national and state-level survey data on annual and monthly scales, regional and city-scale benchmarking datasets on the annual scale, ten regional or city-scale hourly AMI datasets, and multiple anonymous datasets. In particular, the AMI data cover more than 2.3 million meters in residential buildings and more than 0.2 million meters in ~60,000 commercial buildings. Detailed results of calibration and validation can be found in ref.⁸.”

4. How do EnergyPlus models integrated with the statistical model in this study? If EnergyPlus calculates the end-uses and is processed by the statistical model, the fitted data-driven model may not be extrapolated for future climate conditions since it is a data-driven model. This can result in poor quality of the simulation results, and thus the unfounded conclusion.

Thanks for the questions. The outputs from EnergyPlus models are one of the inputs of the statistical model shown in Equation (1). In the revision we clarified this as “where $E_{i,t}^a$ is the aggregated result of EnergyPlus-based end-use load category i at time t ” (Line 573). To help the readers better understand the methods, we also added a new Supplementary Fig. 25 to provide an overview of the bottom-up modeling approach developed in this study. The left half of this figure shows how EnergyPlus-based results and EULP data were ingested into statistical models to produce calibrated hourly site energy use data.

[redacted]

Supplementary Fig. 25. Overview of the hybrid bottom-up urban building energy modeling approach developed in this study and main data sources.

Regarding extrapolation under future climate conditions – good point. Predicting future energy use based on historical energy use–climate relationships and future climate data is a commonly used technique used in existing studies. However, as mentioned by the reviewer, this technique neglects the potential changing relationship between energy use and climate conditions in the future. In this study, we mitigated this issue by introducing future building energy use predictions as the inputs in the calibration model. As a result, the predicted energy use in the 2050s is not solely based on historical energy–climate relationships. Compared with pure statistical (data-driven) models, these physics-based building energy simulations (EnergyPlus here) driven by future weather data can much more realistically reflect the response of energy use to future climate.

Although we do not have real building energy use data in the mid-century, the benefits of using EnergyPlus-based data in calibration have been demonstrated by the good performance during both training and testing periods, as shown in Supplementary Tables 7–9. In comparison, using calibration models without EnergyPlus-based data can yield large discrepancies when the weather patterns (especially air temperature) during the prediction period fall outside the patterns

during model training (i.e., extrapolating). We have observed this issue when using calibration models trained based on 2019 data for predictions in 2018.

As an example, the figure below shows the predicted building electricity use for space cooling during August-September 2018 in Phoenix and Baltimore using Eq. (1) with EnergyPlus-based data (red curves) vs. without EnergyPlus-based data (blue curves), when compared with data from the EULP database (black curves). Note that calibration models here were trained using data in 2019.

Clearly, blue curves are much more consistent with black curves, such as peak values in both urban areas and the low cooling demand period between 09/08/2018 and 09/12/2018 in Baltimore. These comparisons suggest that including EnergyPlus-based data can mitigate the potential extrapolation issue for future predictions.

5. page 20, lines 451-455

“The sum of these individual end-use categories usually exhibits large discrepancies in both the magnitude and load shapes, even on the monthly scale. This is mainly induced by the lack of diverse building characteristics in our prototype-based simulations.”

This statement on discrepancies between prediction and measurement data is partially correct. The discrepancies between model prediction and measurements for buildings are because the prototype models do not represent the actual building design, operation, and control that is being simulated. The lack of diverse building characteristics in prototype-based simulation plays a less important role.

Thanks for pointing out this. We totally agree with the reviewer. By “building characteristics” we meant to include building design, operation, and control, as mentioned in Supplementary Method 5: “The characteristics of these representative building models ... (e.g., location, vintage,

geometry, envelope, equipment, and occupant behavior)". To avoid confusion, we changed "the lack of diverse building characteristics" to "the lack of diverse building characteristics, operations, and controls" in the revised Methods (Line 573).

6. Hourly building energy consumption has been conducted for 277 U.S. cities. However, the analysis of simulation results seems limited. Recommend the author analyze the simulation results in-depth. What can we learn from these substantial modeling efforts?

Thanks for the question and suggestions. We would like to clarify that the hourly building energy consumption is the *foundation* of all analyses conducted in this study. Without hourly building energy simulations, it is not possible to analyze the dynamics of different end-use categories (e.g., space heating vs. cooling) and different energy types (i.e., electricity, natural gas, fuel oil, and propane). In addition, hourly building energy simulations also have two major benefits. First, hourly simulations enable peak energy demand analysis, which can provide critical information for electric grid resilience during warm seasons. Second, these simulations can be used to identify the changes in building energy use during the diurnal cycle. Considering these two benefits, we investigate the peak energy demand as well as diurnal changes in section "Impacts of climate change on seasonal and hourly electricity energy use intensity", and presented results in Fig. 2, Supplementary Figs. 7, 8, 9, 10, and 28. The implications of these analyses for grid capacity expansion and grid resilience are further discussed in the revised Lines 323–335:

"The high temporal resolution of our approach enables detailed assessment across multiple temporal scales. For instance, urban areas in the U.S. are projected to experience diurnal and seasonal changes in total EUI that are much larger than annual mean changes (Fig. 2 and Supplementary Figs. 6–9). These temporally heterogeneous changes should be considered when planning for regional generation capacity expansion and energy storage, especially for a future power grid that relies on variable renewables²⁹. In addition, the substantial increases in the intensity and frequency of summer peak electricity demand in U.S. urban areas (e.g., Supplementary Fig. 10) will require not only a higher grid capacity but also greater resilience against blackouts during extreme heat waves. Recent studies have suggested the rapidly elevated risks of heat-related mortality and morbidity during compound heat wave and grid failure events in several U.S. urban areas^{30,31}. Given the anticipated rise in urban cooling demand across all warming scenarios (Supplementary Fig. 1), improving power grid resilience will become increasingly needed to reduce potential health risks and cope with heat extremes."

REVIEWERS' COMMENTS

Reviewer #1 (Remarks to the Author):

The authors have done a thorough job responding to my comments throughout the manuscript. It reads much better.

I have one critical comment remaining and it goes back to a series of comments in my original review about the balance of communication emphasis between the three drivers considered here. I remain concerned about this. This is mainly a communications issue but an important one.

The abstract offers a good example of this. Three numerical "goalposts" are stated: one about the frequency of peak building electricity EUI under climate change, one on the space cooling/heating EUI under climate change, and one on the total building source energy due to pop growth and power sector "evolutions" (I would probably use "changes").

To me, the emphasis here is unduly focusing on climate change when it is not the largest driver. The use of a maximum metric ("up to" 170%) is also quite different from the mean metric used in number 2 and 3 of the goalposts mentioned. I do think the maximum value of the peak electricity EUI is important and worth emphasizing but it should be partnered with a mean value to provide the reader with the proper balanced context. So, what is the mean change in peak electricity EUI? That is the more balanced metric when placed within an abstract next to "mean" changes in space heating/cooling EUI and "mean" building total source energy change.

I also think the population and power sector changes should be separated in the abstract with their individual impact numbers. That would mirror the title of the paper.

The abstract would be read as perhaps something like:

"Climate, technologies, and socio-economic changes will influence future building energy use in cities. However, current low-resolution regional and state-level analyses are insufficient to reliably assist city-level decision-making. Here we estimate mid-century hourly building energy consumption in 277 U.S. urban areas using a bottom-up approach. Projected future climate change causes heterogeneous changes in energy use intensity (EUI) across urban areas, especially under higher warming scenarios, with a mean XX% change in the frequency of peak building electricity EUI but up to 170% change in some cities. For each 1°C of warming, mean city-scale space-conditioning EUI on average increases/decreases by ~14%/~10% for space cooling/heating. Total building energy consumption changes, however, are mainly driven by population and power sector changes. Across the scenarios considered here, population growth increases mean city-scale source energy use by 7.3% to 52.2% by 2050. Power sector changes decrease mean city-scale source energy use by -8.9% to -10.8%. Climate change results in an increase of 2.1% to 2.5% by 2050. Our findings highlight the need to consider intercity heterogeneity when developing sustainable and resilient urban energy systems."

This seems more representative of the results in the paper.

Reviewer #2 (Remarks to the Author):

The revised paper is recommended for publication.

Responses to Reviewer #1 (Dr. Gurney):

The authors have done a thorough job responding to my comments throughout the manuscript. It reads much better.

I have one critical comment remaining and it goes back to a series of comments in my original review about the balance of communication emphasis between the three drivers considered here. I remain concerned about this. This is mainly a communications issue but an important one.

The abstract offers a good example of this. Three numerical "goalposts" are stated: one about the frequency of peak building electricity EUI under climate change, one on the space cooling/heating EUI under climate change, and one on the total building source energy due to pop growth and power sector "evolutions" (I would probably use "changes").

To me, the emphasis here is unduly focusing on climate change when it is not the largest driver. The use of a maximum metric ("up to" 170%) is also quite different from the mean metric used in number 2 and 3 of the goalposts mentioned. I do think the maximum value of the peak electricity EUI is important and worth emphasizing but it should be partnered with a mean value to provide the reader with the proper balanced context. So, what is the mean change in peak electricity EUI? That is the more balanced metric when placed within an abstract next to "mean" changes in space heating/cooling EUI and "mean" building total source energy change.

I also think the population and power sector changes should be separated in the abstract with their individual impact numbers. That would mirror the title of the paper.

The abstract would be read as perhaps something like:

“Climate, technologies, and socio-economic changes will influence future building energy use in cities. However, current low-resolution regional and state-level analyses are insufficient to reliably assist city-level decision-making. Here we estimate mid-century hourly building energy consumption in 277 U.S. urban areas using a bottom-up approach. Projected future climate change causes heterogeneous changes in energy use intensity (EUI) across urban areas, especially under higher warming scenarios, with a mean XX% change in the frequency of peak building electricity EUI but up to 170% change in some cities. For each 1°C of warming, mean city-scale space-conditioning EUI on average increases/decreases by ~14%/~10% for space cooling/heating. Total building energy consumption changes, however, are mainly driven by population and power sector changes. Across the scenarios considered here, population growth increases mean city-scale source energy use by 7.3% to 52.2% by 2050. Power sector changes decrease mean city-scale source energy use by -8.9% to -10.8%. Climate change results in an increase of 2.1% to 2.5% by 2050. Our findings highlight the need to consider intercity heterogeneity when developing sustainable and resilient urban energy systems.”

This seems more representative of the results in the paper.

Thank you for pointing this out. Following your suggestion, we have revised the abstract by adding the average changes of peak EUI frequencies and the individual impact numbers of

climate change, population growth, and power sector decarbonization. The revised abstract is copied below:

“Climate, technologies, and socio-economic changes will influence future building energy use in cities. However, current low-resolution regional and state-level analyses are insufficient to reliably assist city-level decision-making. Here we estimate mid-century hourly building energy consumption in 277 U.S. urban areas using a bottom-up approach. The projected future climate change results in heterogeneous changes in energy use intensity (EUI) among urban areas, particularly under higher warming scenarios, with on average 10.1–37.7% increases in the frequency of peak building electricity EUI but over 110% increases in some cities. For each 1°C of warming, the mean city-scale space-conditioning EUI experiences an average increase/decrease of ~14%/~10% for space cooling/heating. However, heterogeneous city-scale building source energy use changes are primarily driven by population and power sector changes, on average ranging from –9% to 40% with consistent south–north gradients under different scenarios. Across the scenarios considered here, the changes in city-scale building source energy use, when averaged over all urban areas, are as follows: –2.5% to –2.0% due to climate change, 7.3% to 52.2% due to population growth, and –17.1% to –8.9% due to power sector decarbonization. Our findings underscore the necessity of considering intercity heterogeneity when developing sustainable and resilient urban energy systems.”

Responses to Reviewer #2:

The revised paper is recommended for publication.

Thanks again for all your insightful and constructive comments.